# Kernel Memory Networks:
# A Unifying Framework for Memory Modeling

**Georgios Iatropoulos**[1,2]    **Johanni Brea**[1,*]    **Wulfram Gerstner**[1,*]

`{firstname.lastname}@epfl.ch`
[1]Laboratory of Computational Neuroscience, [2]Blue Brain Project
École Polytechnique Fédérale de Lausanne
Switzerland

## Abstract

We consider the problem of training a neural network to store a set of patterns with maximal noise robustness. A solution, in terms of optimal weights and state update rules, is derived by training each individual neuron to perform either kernel classification or interpolation with a minimum weight norm. By applying this method to feed-forward and recurrent networks, we derive optimal models, termed kernel memory networks, that include, as special cases, many of the hetero- and auto-associative memory models that have been proposed over the past years, such as modern Hopfield networks and Kanerva's sparse distributed memory. We modify Kanerva's model and demonstrate a simple way to design a kernel memory network that can store an exponential number of continuous-valued patterns with a finite basin of attraction. The framework of kernel memory networks offers a simple and intuitive way to understand the storage capacity of previous memory models, and allows for new biological interpretations in terms of dendritic non-linearities and synaptic cross-talk.

## 1   Introduction

Although the classical work on attractor neural networks reached its peak in the late 1980's, with the publication of a number of seminal works [e.g., 2, 20, 22, 26], recent years have seen a renewed interest in the topic, motivated by the popularity of the attention mechanism [65], external memory-augmented neural networks [24, 66], as well as a new generation of energy-based attractor networks models, termed modern Hopfield networks (MHNs), capable of vastly increased memory storage [17, 35]. Recent efforts to understand the theoretical foundation of the attention mechanism have, in fact, shown that it can be linked to Hopfield networks [36, 57], but also to Kanerva's sparse distributed memory (SDM) [8, 30], and to the field of kernel machines [63, 68]. The last connection is particularly intriguing, in light of the many theoretical commonalities between neural networks and kernel methods [10, 11, 28, 47, 67]. Overall, these results suggest that a unified view can offer new insights into memory modeling and new tools for leveraging memory in machine learning.

In this work, we aim to clarify some of the overlap between the fields of memory modeling and statistical learning, by integrating and formalizing a set of theoretical connections between Hopfield networks, the SDM, kernel machines, and neuron models with non-linear dendritic processing.

---

[*]Joint senior authors.

36th Conference on Neural Information Processing Systems (NeurIPS 2022).

## 1.1 Our contribution

- We derive a set of normative kernel-based models that describe the general mathematical structure of feed-forward (i.e., hetero-associative) and recurrent (i.e., auto-associative) memory networks that can perform error-free recall of a given set of patterns with maximal robustness to noise.

- We show that the normative models include, as special cases, the classical and modern Hopfield network, as well as the SDM.

- We derive a simple attractor network model for storing an exponential number of continuous-valued patterns with a finite basin of attraction. We discuss its similarity to attention.

Furthermore, we explain how classifiers with non-linear kernels can be interpreted as general forms of neuron models with non-linear dendritic activation functions and synaptic cross-talk.

## 1.2 Related work

Our work is primarily related to [8, 9, 35, 36, 45, 57]. While MHNs are extensively analyzed in [35, 36, 45, 57], the approach is energy-based and makes no statements about the relation between MHNs and kernel methods; a brief comment in [57] mentions some similarity to SVMs, but this is not further explained. The work by [8] focuses on the SDM and its connection to attention. It observes that the classical Hopfield network is a special case of the SDM, but no further generalization is made, and kernel methods are not mentioned. In our work, we place MHNs and the SDM in a broader theoretical context by showing that *both* models are special suboptimal cases of a family of memory networks that can be derived with a normative kernel-based approach.

## 2 Background

Consider the following simple model of hetero-associative memory: a single-layer feed-forward network consisting of a single output neuron connected to $N_{\text{in}}$ inputs with the weights $\mathbf{w} \in \mathbb{R}^{N_\phi}$. The output $s_{\text{out}} \in \{\pm 1\}$ is given by

$$s_{\text{out}} = \text{sgn}\left[\mathbf{w}^\top \boldsymbol{\phi}(\mathbf{s}_{\text{in}}) - \theta\right] \tag{1}$$

where $\mathbf{s}_{\text{in}}$ is the input vector (also called query), $\theta$ the threshold, and $\phi$ a function that maps the "raw" input to a $N_\phi$-dimensional feature space, where typically $N_\phi \gg N_{\text{in}}$. Suppose that we are given a set of $M$ input-output patterns $\{\boldsymbol{\xi}_{\text{in}}^\mu, \xi_{\text{out}}^\mu\}_{\mu=1}^M$, in which every entry $\xi$ is randomly drawn from $\{\pm 1\}$ with sparseness $f := \mathbb{P}(\xi = 1)$. In order for the neuron to store the patterns in a way that maximizes the amount of noise it can tolerate while still being able to recall all patterns without errors, one needs to find the weights that produce the output $\xi_{\text{out}}^\mu$ in response to the input $\boldsymbol{\xi}_{\text{in}}^\mu$, $\forall \mu$, and that maximize the smallest Euclidean distance between the inputs and the neuron's decision boundary. Using Gardner's formalism [20, 22], this problem can be expressed as

$$\underset{\mathbf{w}}{\arg\max} \ \kappa \quad \text{s.t.} \quad \xi_{\text{out}}^\mu\left(\mathbf{w}^\top \boldsymbol{\phi}(\boldsymbol{\xi}_{\text{in}}^\mu) - \theta\right) \geq \kappa, \ \forall \mu \\ \|\mathbf{w}\|_2 = \bar{w} \tag{2}$$

where $\bar{w} > 0$ is a constant. This is equivalent to solving

$$\underset{\mathbf{w}}{\min} \ \|\mathbf{w}\|_2 \quad \text{s.t.} \quad \xi_{\text{out}}^\mu\left(\mathbf{w}^\top \boldsymbol{\phi}(\boldsymbol{\xi}_{\text{in}}^\mu) - \theta\right) \geq 1, \ \forall \mu \tag{3}$$

which can be directly identified as the support vector machine (SVM) problem for separable data [13]. The solution to Eq. 3 can today be found in any textbook on basic machine learning methods, and yields an optimal output rule that can be written in a *feature* and *kernel* form

$$s_{\text{out}} = \text{sgn}\left[\sum_\mu^M \alpha^\mu \xi_{\text{out}}^\mu \boldsymbol{\phi}(\boldsymbol{\xi}_{\text{in}}^\mu)^\top \boldsymbol{\phi}(\mathbf{s}_{\text{in}}) - \theta\right] = \text{sgn}\left[\sum_\mu^M \alpha^\mu \xi_{\text{out}}^\mu K(\boldsymbol{\xi}_{\text{in}}^\mu, \mathbf{s}_{\text{in}}) - \theta\right] \tag{4}$$

where we, in the latter expression, have used the "kernel-trick" $K(\mathbf{x}_i, \mathbf{x}_j) = \boldsymbol{\phi}(\mathbf{x}_i)^\top \boldsymbol{\phi}(\mathbf{x}_j)$. The solution depends on the Lagrange coefficients $\alpha^\mu \geq 0$, many of which are typically zero. Patterns with $\alpha^\mu > 0$ are called *support vectors*.

# 3 Kernel memory networks for binary patterns

## 3.1 *Hetero*-associative memory as a feed-forward SVM network

We begin by considering a hetero-associative memory network with an arbitrary number $N_{\text{out}}$ output neurons, whose combined state we denote $\mathbf{s}_{\text{out}}$. In order for the network as a whole to be able to tolerate a maximal level of noise and still successfully recall its stored memories, we solve Eq. 3 for each neuron independently. As each neuron can have a different classification boundary along with a different set of support vectors, its weights will, in general, be characterized by an independent set of $M$ Lagrange coefficients. To simplify the notation, we represent these coefficients $\alpha_i^\mu$, across neurons $i$ and patterns $\mu$, as entries in the matrix $\mathbf{A}$, where $(\mathbf{A})_{i\mu} = \alpha_i^\mu$. We also combine all thresholds in the vector $\boldsymbol{\theta} = (\theta_1, \ldots, \theta_{N_{\text{out}}})$, and all input and output patterns as columns in the matrices $\mathbf{X}_{\text{in}} = (\boldsymbol{\xi}_{\text{in}}^1, \ldots, \boldsymbol{\xi}_{\text{in}}^M)$ and $\mathbf{X}_{\text{out}} = (\boldsymbol{\xi}_{\text{out}}^1, \ldots, \boldsymbol{\xi}_{\text{out}}^M)$. Finally, we assume that all neurons have the same feature map, so that $\phi_i = \phi, \forall i$ (see Fig. 1). All functions are applied column-wise when the argument is a matrix, for example $\phi(\mathbf{X}_{\text{in}}) = (\phi(\boldsymbol{\xi}_{\text{in}}^1), \ldots, \phi(\boldsymbol{\xi}_{\text{in}}^M))$. The optimal response of the network can now be compactly summarized as follows.

**Property 1** (Robust hetero-associative memory network). *A single-layer hetero-associative memory network trained to recall the patterns $\mathbf{X}_{\text{out}}$ in response to the inputs $\mathbf{X}_{\text{in}}$ with maximal noise robustness, has an optimal output rule that can be written as*

$$\mathbf{s}_{\text{out}} = \text{sgn}\Big[(\mathbf{A} \odot \mathbf{X}_{\text{out}})\phi(\mathbf{X}_{\text{in}})^\top \phi(\mathbf{s}_{\text{in}}) - \boldsymbol{\theta}\Big] \qquad \textit{(feature form)} \qquad (5)$$

$$= \text{sgn}\big[(\mathbf{A} \odot \mathbf{X}_{\text{out}})K(\mathbf{X}_{\text{in}}, \mathbf{s}_{\text{in}}) - \boldsymbol{\theta}\big] \qquad \textit{(kernel form)} \qquad (6)$$

*where $\odot$ denotes the Hadamard product.*

## 3.2 *Auto*-associative memory as a recurrent SVM network

The hetero-associative network can be made auto-associative by setting $N_{\text{out}} = N_{\text{in}}$ and $\mathbf{X}_{\text{out}} = \mathbf{X}_{\text{in}}$. The network is now effectively recurrent, as each neuron can serve both as an input and output simultaneously (see Fig. 1). Consider a recurrent network with $N$ neurons, whose state at time point $t$ is denoted $\mathbf{s}^{(t)} \in \{\pm 1\}^N$, and whose dynamics evolve according to the update rule

$$s_i^{(t+1)} = \text{sgn}\Big[\mathbf{w}_i^\top \phi(\mathbf{s}^{(t)}) - \theta_i\Big] \qquad (7)$$

where $\mathbf{w}_i \in \mathbb{R}^{N_\phi}$ is the weight vector to neuron $i = 1, \ldots, N$. In order to make the patterns $\{\boldsymbol{\xi}^\mu\}_{\mu=1}^M$ fixed points of the network dynamics, we train each neuron $i$ independently on every pattern $\mu$ to, again, produce the response $\xi_i^\mu$ when the rest of the network is initialized in $\boldsymbol{\xi}^\mu$. Moreover, we maximize the amount of noise that can be tolerated by the network while maintaining error-free recall by maximizing the smallest Euclidean distance between each neuron's decision boundary and its inputs. This maximizes the size of the attractor basins [18, 32]. The problem of training the entire network is, in this way, transformed into the problem of training $N$ separate classifiers according to

$$\min_{\mathbf{w}_i} \|\mathbf{w}_i\|_2 \quad \text{s.t.} \quad \xi_i^\mu\Big(\mathbf{w}_i^\top \phi(\boldsymbol{\xi}^\mu) - \theta_i\Big) \geq 1, \ \forall \mu, i \ . \qquad (8)$$

The solution can be obtained by slightly modifying Property 1, and is stated below.

**Property 2.1** (Robust auto-associative memory). *A recurrent auto-associative memory network trained to recall the patterns $\mathbf{X}$ with maximal noise robustness has an optimal synchronous update rule that can be written as*

$$\mathbf{s}^{(t+1)} = \text{sgn}\Big[(\mathbf{A} \odot \mathbf{X})\phi(\mathbf{X})^\top \phi(\mathbf{s}^{(t)}) - \boldsymbol{\theta}\Big] \qquad \textit{(feature form)} \qquad (9)$$

$$= \text{sgn}\Big[(\mathbf{A} \odot \mathbf{X})K(\mathbf{X}, \mathbf{s}^{(t)}) - \boldsymbol{\theta}\Big] \qquad \textit{(kernel form)} \qquad (10)$$

*Remark.* With a linear feature map $\phi(\mathbf{x}) = \mathbf{x}$, the optimal update is reduced to

$$\mathbf{s}^{(t+1)} = \text{sgn}\Big[(\mathbf{A} \odot \mathbf{X})\mathbf{X}^\top \mathbf{s}^{(t)} - \boldsymbol{\theta}\Big] \qquad (11)$$

where $(\mathbf{A} \odot \mathbf{X})\mathbf{X}^\top$ can be identified as the general form of the optimal weight matrix.

The solution described by Property 2.1 does not, in general, prohibit a neuron from having self-connections. Applying this constraint yields the following result.

**Property 2.2** (Robust auto-associative memory without self-connections). *A recurrent auto-associative memory network without self-connections, with the inner-product kernel $K(\mathbf{x}_i, \mathbf{x}_j) = k(\mathbf{x}_i^\top \mathbf{x}_j)$, that has been trained to recall the patterns $\mathbf{X}$ with maximal noise robustness, has an optimal asynchronous update rule that can be written in the kernel form*

$$s_i^{(t+1)} = \mathrm{sgn}\left[\sum_\mu^M \alpha_i^\mu \xi_i^\mu k\left(\sum_{j\neq i}^N \xi_j^\mu s_j^{(t)}\right) - \theta_i\right]. \tag{12}$$

**Storage capacity.** An intuition for the storage capacity scaling of the hetero- and auto-associative memory networks can be gained by observing that the network as a whole will be able to successfully recall patterns as long as each neuron is able to correctly classify its inputs (or is very unlikely to produce an error). The capacity of the network can thereby be derived from the capacity of each individual neuron. It is well-known that a linear binary classifier can learn to correctly discriminate a maximum of $M_{\max} \approx 2D_{\mathrm{VC}}$ random patterns, where $D_{\mathrm{VC}}$ is the Vapnik-Chervonenkis dimension of the classifier [15, 20, 40, ch. 40]. For a neuron with $N$ inputs and a linear feature map $\phi(\mathbf{x}) = \mathbf{x}$, this results in $D_{\mathrm{VC}} = N$ and, thus, the capacity $M_{\max} \approx 2N$. Suppose, on the other hand, that the kernel is a homogeneous polynomial of degree $p$, so that $K(\mathbf{x}_i, \mathbf{x}_j) = (\mathbf{x}_i^\top \mathbf{x}_j)^p$. In this case, $\phi$ will contain all monomials of degree $p$ composed of the entries in $\mathbf{x}$. As there are $\mathcal{O}(N^p)$ unique $p$-degree monomials (see Appendix A.1), the input dimensionality and $M_{\max}$ will be $\mathcal{O}(N^p)$. For the exponential kernel, which we can write as $K(\mathbf{x}_i, \mathbf{x}_j) = \exp(\mathbf{x}_i^\top \mathbf{x}_j) = \sum_{p=0}^\infty (\mathbf{x}_i^\top \mathbf{x}_j)^p/p!$, the dimensionality of $\phi$ will be $\sum_{p=0}^N \binom{N}{p} = 2^N$, which yields $M_{\max} \sim \mathcal{O}(e^N)$.

**Special cases.** In the following sections, we will show that many of the models of hetero- and auto-associative memory that have been proposed over the past years are special cases of the solutions in Properties 1, 2.1, and 2.2, characterized by specific choices of $\mathbf{A}$, $\phi$, and $K$.

## 3.3 Kanerva's sparse distributed memory is a feed-forward SVM network

The sparse distributed memory (SDM), developed by Kanerva [30], is one of the most famous examples of a hetero-associative memory model. It has lately received much attention in the context of generative memory models [69] and attention layers in transformers [8].

The SDM consists of a register of $N_\phi$ memory slots, each associated with an address $\mathbf{z}_i \in \{\pm 1\}^{N_{\mathrm{in}}}$, $i = 1, \ldots, N_\phi$. All addresses are listed as rows in the matrix $\mathbf{Z} = (\mathbf{z}_1, \ldots, \mathbf{z}_{N_\phi})^\top$. The content of each slot is represented by an $N_{\mathrm{out}}$-dimensional vector, initialized at zero. Suppose that we wish to store the $M$ patterns $\mathbf{X}_{\mathrm{out}} = (\boldsymbol{\xi}_{\mathrm{out}}^1, \ldots, \boldsymbol{\xi}_{\mathrm{out}}^M)$ in the addresses $\mathbf{X}_{\mathrm{in}} = (\boldsymbol{\xi}_{\mathrm{in}}^1, \ldots, \boldsymbol{\xi}_{\mathrm{in}}^M)$, where all entries are random and bipolar. The basic idea of the SDM is to write the data to, and later read it from, multiple memory slots at once (hence the distributed storage); this ensures a degree of noise-robustness. In mathematical terms, the read-out of the SDM provided with a query $\mathbf{s}_{\mathrm{in}}$, is given by

$$\mathbf{s}_{\mathrm{out}} = \mathrm{sgn}\left[\mathbf{X}_{\mathrm{out}}\,\Theta(\mathbf{Z}\mathbf{X}_{\mathrm{in}} - b)^\top\,\Theta(\mathbf{Z}\mathbf{s}_{\mathrm{in}} - b)\right] \tag{13}$$

where $\Theta$ the Heaviside function with bias $b = N_{\mathrm{in}} - 2r$, and $r$ is a parameter that determines the precision of the writing and reading process. Upon comparing Eqs. 13 and 5, the SDM can be directly identified as a special case of a suboptimal feed-forward SVM network in the feature form, with $\mathbf{A} = \mathbf{1}$, $\boldsymbol{\theta} = \mathbf{0}$, and the feature map $\phi_{\mathrm{SDM}}(\mathbf{x}) = \Theta(\mathbf{Z}\mathbf{x} - b)$. When viewed as a kernel method, the function of the SDM is to store the dense addresses $\mathbf{X}_{\mathrm{in}}$ as sparse high-dimensional representations $\phi_{\mathrm{SDM}}$, to make it easier to later determine the slots closest to a query $\mathbf{s}_{\mathrm{in}}$, and retrieve the relevant data.

**Capacity.** As the SDM is linear in $\phi_{\mathrm{SDM}}$, with $D_{\mathrm{VC}} \approx N_\phi$, it follows from the analysis in Sec. 3.2 that one should expect the capacity to scale as $M_{\max} \sim \mathcal{O}(N_\phi)$. Moreover, one should expect a proportionality constant $\sim 0.1$, since the SDM is suboptimal relative to the feed-forward SVM network, analogously to how the classical Hopfield network is suboptimal relative to the recurrent SVM network (see Sec. 3.4). This is consistent with earlier proofs [12, 31].

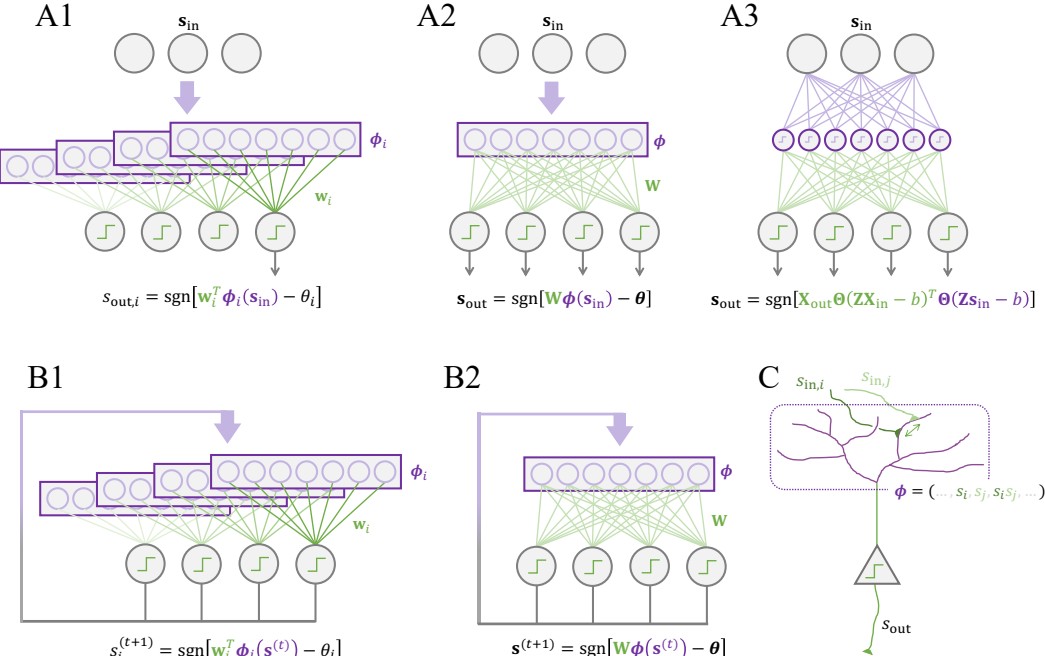

Figure 1: Graphical representation of (A1-A2) the feed-forward SVM network, (A3) the SDM, (B1-B2) the recurrent SVM network, and (C) an SVM mapped to the anatomy of a pyramidal cell (see Sec. 5). Circles represent neurons, while boxes represent the input transformation by the feature map $\phi$, which can be dependent (A1, B1) or independent (A2, B2) of neuron index $i$.

**Kernel of an infinite SDM.** In practice, an SDM with a large number of memory slots $N_\phi$ requires calculations involving a large address matrix $\mathbf{Z}$. This can be avoided by applying the kernel-trick to Eq. 13 in the limit $N_\phi \to \infty$, which allows for the output to be computed with

$$\mathbf{s}_{\text{out}} = \text{sgn}\big[\mathbf{X}_{\text{out}} K_{\text{SDM}}(\mathbf{X}_{\text{in}}, \mathbf{s}_{\text{in}})\big] \tag{14}$$

where we have defined the kernel as

$$K_{\text{SDM}}(\mathbf{x}_i, \mathbf{x}_j) = \lim_{N_\phi \to \infty} \frac{\phi_{\text{SDM}}(\mathbf{x}_i)^\top \phi_{\text{SDM}}(\mathbf{x}_j)}{N_\phi} \tag{15}$$

in order to ensure convergence. In this section, we will derive this kernel for two different variants of the SDM and demonstrate that both are translation-invariant. It is interesting to note here that $\phi_{\text{SDM}}$ is equivalent to a single-layer neural network with $N_\phi$ neurons, weights $\mathbf{Z}$, and bias $b$. This means that $K_{\text{SDM}}$ is equivalent to the kernel of an infinitely wide neural network [11, 47, 67].

We begin by noticing that $\phi_{\text{SDM}}(\mathbf{x})$ has a geometrical interpretation [8, 31]. It is a binary vector that indicates those memory addresses in $\mathbf{Z}$ that differ by at most $r$ bits compared to $\mathbf{x}$. For any two bipolar vectors $\mathbf{z}$ and $\mathbf{x}$, the bit-wise difference can be computed as $\frac{1}{2}|\mathbf{z} - \mathbf{x}| = \frac{1}{4}\|\mathbf{z} - \mathbf{x}\|_2^2$. This means that $\phi_{\text{SDM}}(\mathbf{x})$ indicates all addresses that lie within a sphere centered at $\mathbf{x}$ with radius $2\sqrt{r}$. Consequently, the inner product $\phi_{\text{SDM}}(\mathbf{x}_i)^\top \phi_{\text{SDM}}(\mathbf{x}_j)$ is the number of addresses located in the overlapping volume of two spheres centered at $\mathbf{x}_i$ and $\mathbf{x}_j$. Although an exact calculation of this quantity can be found in [8, 30], its connection to the SDM kernel has, to the best of our knowledge, not previously been made. We therefore modify the previously published expression with a normalization factor $1/2^{N_{\text{in}}}$ and state the following property.

**Property 3.1** (Kernel of an infinite SDM on the hypercube). *In the limit $N_\phi \to \infty$, the kernel of an SDM with $N_\phi$ memory slots, whose addresses are randomly drawn from $\{\pm 1\}^{N_{\text{in}}}$, is given by*

$$K_{\text{SDM}}(\mathbf{x}_i, \mathbf{x}_j) = \frac{1}{2^{N_{\text{in}}}} \sum_{i=N_{\text{in}}-r-\lfloor\frac{\Delta}{2}\rfloor}^{N_{\text{in}}-\Delta} \sum_{j=[N_{\text{in}}-r-i]_+}^{\Delta-(N_{\text{in}}-r-i)} \binom{N_{\text{in}} - \Delta}{i} \cdot \binom{\Delta}{j} \tag{16}$$

*where $r$ is the bit-wise error threshold and $\Delta$ is the bit-wise difference between $\mathbf{x}_i$ and $\mathbf{x}_j$, given by $\Delta = \frac{1}{2}|\mathbf{x}_i - \mathbf{x}_j| = \frac{1}{4}\|\mathbf{x}_i - \mathbf{x}_j\|_2^2$.*

The SDM can also be implemented with continuous addresses, randomly placed on a unit hypersphere of $(N_{\text{in}} - 1)$ dimensions, denoted $\mathbb{S}^{N_{\text{in}} - 1}$. The vector $\boldsymbol{\phi}_{\text{SDM}}(\mathbf{x})$ now indicates all addresses that lie within a hyperspherical cap centered at $\mathbf{x}$ with an angle $\arccos(b)$ between its central axis and the rim. The inner product $\boldsymbol{\phi}_{\text{SDM}}(\mathbf{x}_i)^\top \boldsymbol{\phi}_{\text{SDM}}(\mathbf{x}_j)$ is the number of addresses located in the overlapping area of two spherical caps centered at $\mathbf{x}_i$ and $\mathbf{x}_j$. While a calculation of this quantity, again, can be found in [8], it has not previously been connected to the kernel of an SDM. We simplify the previously published result and also derive a closed-form approximation, valid for highly sparse $\boldsymbol{\phi}_{\text{SDM}}$ (see Appendix B for details). The results are summarized below.

**Property 3.2** (Kernel of an infinite SDM on the hypersphere). *In the limit $N_\phi \to \infty$, the kernel of an SDM with $N_\phi$ memory slots, whose addresses are randomly drawn from $\mathbb{S}^{N_{\text{in}} - 1}$, is given by*

$$K_{\text{SDM}}(\mathbf{x}_i, \mathbf{x}_j) = \frac{N_{\text{in}} - 2}{2\pi} \int_{\alpha_x}^{\alpha_b} \sin(\varphi)^{N_{\text{in}} - 2} B\left[1 - \frac{\tan^2(\alpha_x)}{\tan^2(\varphi)}; \frac{N_{\text{in}} - 2}{2}, \frac{1}{2}\right] \mathrm{d}\varphi \qquad (17)$$

*where $\alpha_x = \frac{1}{2}\arccos(\mathbf{x}_i^\top \mathbf{x}_j)$, $\alpha_b = \arccos(b)$, and $B$ is the incomplete Beta function. In the highly sparse regime, when $0.9 \lesssim b < 1$ and $\frac{1}{N_\phi}\|\boldsymbol{\phi}_{\text{SDM}}\|_0 \ll 1$, the kernel can be approximated with*

$$K_{\text{SDM}}(\mathbf{x}_i, \mathbf{x}_j) \approx \frac{\hat{b}^{N_{\text{in}} - 1}}{2\pi} B\left[1 - \left(\frac{\Delta}{\hat{b}}\right)^2; \frac{N_{\text{in}}}{2}, \frac{1}{2}\right] \qquad (18)$$

*where $\Delta = \frac{1}{2}\|\mathbf{x}_i - \mathbf{x}_j\|_2$ and $\hat{b} = \sin(\arccos(b))$.*

In conclusion, an infinitely large SDM with sparse internal representations $\boldsymbol{\phi}_{\text{SDM}}$, can be represented as a suboptimal case of a feed-forward SVM network with a translation-invariant kernel.

### 3.4 The modern Hopfield network is a recurrent SVM network

The Hopfield network [26] is, arguably, the most well-known model of auto-associative memory. In its modern form [35], it is a recurrent network of $N$ neurons with the state $\mathbf{s}^{(t)}$, whose dynamics are governed by the energy and state update rule

$$E = -\sum_\mu^M F\left(\sum_i^N \xi_i^\mu s_i^{(t)}\right), \quad s_i^{(t+1)} = \text{sgn}\left[\sum_\mu^M \xi_i^\mu F'\left(\sum_{j \neq i}^N \xi_j^\mu s_j^{(t)}\right)\right] \qquad (19)$$

where $F$ is a smooth function, typically a sigmoid, polynomial, or exponential. This "generalized" Hopfield model has a long history [see, e.g., 1, 21, 25, 37] but has received renewed attention in recent years under the name *modern Hopfield network* (MHN) or *dense associative memory* [17, 35]. By comparing Eq. 19 with Eq. 12, the state update of the MHN can be identified as a special case of a suboptimal recurrent SVM network in the kernel form, with $k = F'$, $\mathbf{A} = \mathbf{1}$, and $\boldsymbol{\theta} = \mathbf{0}$ (since $f = 0.5$). With a linear $F'(x) = x$, the MHN reduces to the classical Hopfield network, which is a special case of the recurrent SVM network with the linear kernel $k(\mathbf{x}_i^\top \mathbf{x}_j) = \mathbf{x}_i^\top \mathbf{x}_j$.

**Capacity.** The storage capacity of the MHN has been shown to depend on the shape of $F'$. In the linear case, the capacity is famously limited to $\sim 0.1N$ patterns, depending on the precision of retrieval [2, 42]. If, on the other hand, $F'$ is polynomial with degree $p$, the capacity scales as $M_{\text{max}} \sim \mathcal{O}(N^p)$ [35], while an exponential $F'$ endows the network with a capacity $M_{\text{max}} \sim \mathcal{O}(e^N)$ [17]. From the perspective of the kernel memory framework, this scaling directly follows from the analysis in Sec. 3.2 with $k = F'$. In fact, in the regime of low errors, the kernel memory framework can also be used to derive a more precise capacity scaling for the classical Hopfield network. We first note that any one-shot learning rule that implies $\mathbf{A} > 0$ is equivalent to an SVM network where every stored pattern is a support vector. Such a heuristic is only likely to be close to the optimal solution and perform well in large networks with very few patterns, as high-dimensional linear SVMs trained on few patterns are highly likely to find solutions where all patterns are support vectors; this effect has been termed *support vector proliferation* [4]. Restricting the network to this regime limits the capacity to $M_{\text{max}} \sim \mathcal{O}(\frac{N}{2 \log N})$, consistent with the result in [42] (see Appendix A.2).

**Iterative learning rules.** The problem of iteratively training MHNs with biologically plausible online learning rules has recently been studied [64], with a resulting storage capacity ranging from $\sim 0.16N$ to $\sim N$, depending on the exact implementation. The aim, in general, of such studies is to find a learning rule capable of producing a capacity close to the theoretical maximum $\sim 2N$. For this purpose, the perspective of kernel memory networks can be particularly helpful, as many of the algorithms that have been developed over the past two decades to optimize SVMs can be utilized for MHNs as well. For example, a network formulated in the feature form can be trained with the stochastic batch perceptron rule [14, 34], the passive aggressive rules [16], the minnorm rule [5], as well as with likelihood maximization applied to logistic regression [29, 46, 59]. In the kernel form, two of the most well-known online algorithms for training linear and non-linear SVMs are the Adatron [3] and the Kernel-Adatron [19]. A performance comparison between iterative learning and the modern Hopfield learning rule can be found in Appendix C.

**Generalization.** Viewing the MHN as a recurrent network of SVMs can also facilitate a more intuitive understanding of its ability to generalize, when used as a conventional classifier. In this setting, one designates a subset of the neurons as input units, and the remaining neurons as outputs. Given a set of input-output associations, one optimizes the memory patterns $\boldsymbol{\xi}^\mu$ using, for example, gradient descent. Such an experiment was performed by Krotov and Hopfield [35] on the MNIST data set, using a polynomial non-linearity $F(x) = x^p$. Results showed that the test error first improved as $p$ increased from 2 to 3, but later deteriorated for high degrees, like $p = 20$. While it may be difficult to explain this behavior within an energy-based framework, it is entirely expected when viewed from the SVM perspective: a kernel of low polynomial degree has too few degrees of freedom to fit the classification boundary in the training set, causing *underfitting*, while a polynomial of too high degree grants the model too much flexibility, which results in *overfitting*.

**The pseudoinverse learning rule.** The coefficients in $\mathbf{A}$ are, in general, computed numerically, and cannot be written in closed form. However, in the special case when Eq. 8 is underdetermined, meaning $M < N_\phi$, a closed-form (but suboptimal) solution can be obtained using the *least-squares SVM* method [60]. The result is a generalized form of the *pseudoinverse learning rule* [50]. See Appendix D for details.

## 4   Kernel memory networks for continuous patterns

### 4.1   *Auto*-associative memory as a recurrent interpolation network

So far, we have considered memory models designed to store only bipolar patterns. We now relax this constraint and allow patterns to be continuous-valued. We first observe that any set of patterns $\mathbf{X} \in \mathbb{R}^{N \times M}$ can be made fixed points of the dynamics by training each neuron $i$ to interpolate $\xi_i^\mu$ when the rest of the network is initialized in $\boldsymbol{\xi}^\mu$, for every pattern $\mu$. Assuming that the model is equipped with a kernel that allows for each fixed point to also be attracting, we can ensure that a lower bounding estimate of the size of the attractor basin is maximized by finding the interpolation with minimum weight norm (see Appendix E.1 for proof). These results are summarized below.

**Property 4** (Robust auto-associative memory with continuous patterns)**.** *Suppose that the dynamics of a recurrent auto-associative memory network evolve according to the synchronous update rule*

$$\mathbf{s}^{(t+1)} = \mathbf{X}\mathbf{K}^\dagger K(\mathbf{X}, \mathbf{s}^{(t)}) \tag{20}$$

*where $\mathbf{K} = K(\mathbf{X}, \mathbf{X}) = \phi(\mathbf{X})^\top \phi(\mathbf{X})$ is the kernel matrix and $\mathbf{K}^\dagger$ its Moore-Penrose pseudoinverse, where $\mathbf{K}^\dagger = \mathbf{K}^{-1}$ if $\phi(\mathbf{X})$ is full column rank. Then, the dynamics of the network is guaranteed to have the fixed points $\mathbf{X}$. Moreover, if the points are attracting, Eq. 20 maximizes a lower bound of the attractor basin sizes.*

### 4.2   A recurrent interpolation network with exponential capacity

Memory models for continuous data [e.g., 27, 33, 48] have generally received less attention than their binary counterparts. Recently, however, Ramsauer et al. [57] proposed an energy-based model capable of storing an exponential number of continuous-valued patterns (we will refer to this model as the softmax network). While the structure of this model is similar to Eq. 20, it cannot be analyzed within

the framework of Property 4, as it involves a kernel that is neither symmetric nor positive-definite [68].

Nonetheless, we will in this section demonstrate that it is possible to use conventional kernel methods to design an attractor network with exponential capacity for continuous patterns. We utilize the properties of the SDM by using a translation-invariant kernel with a fixed spatial scale $r$. For the sake of simplicity, we choose the exponential power kernel ($\text{Exp}_\beta$)

$$K_{\exp_\beta}(\mathbf{x}_i, \mathbf{x}_j) = \exp\left[-\left(\frac{1}{r}\|\mathbf{x}_i - \mathbf{x}_j\|_2\right)^\beta\right] \tag{21}$$

where $\beta, r > 0$. These parameters determine the shape of the attractor basin that surrounds each pattern. While $r$ roughly sets the radius of attraction, $\beta$ represents an inverse temperature which changes the steepness of the boundary of the attractor basin. Moreover, as long as the patterns are unique, the kernel matrix is invertible and we have $\mathbf{K}_{\exp_\beta}^\dagger = \mathbf{K}_{\exp_\beta}^{-1}$ [44].

We will now analyze the noise robustness and storage capacity of this model. To make the analysis tractable, we will operate in the regime of low temperatures, meaning the limit $\beta \to \infty$. We first establish the following three properties.

**Property 5.1** (The $\text{Exp}_\beta$ network at zero temperature). *Given a set of unique patterns $\{\boldsymbol{\xi}^\mu\}_{\mu=1}^M$ with $\min_{\mu,\nu\neq\mu}\|\boldsymbol{\xi}^\mu - \boldsymbol{\xi}^\nu\|_2 > r$, the state update rule for the $\text{Exp}_\beta$ network at $\beta \to \infty$ reduces to*

$$\mathbf{s}^{(t+1)} = \mathbf{X}\,\Theta(r - \|\mathbf{X} - \mathbf{s}^{(t)}\|_2) \tag{22}$$

*where $\Theta(\cdot)$ is the Heaviside function with $\Theta(0) = e^{-1}$ (see Appendix E.2.1).*

*Remark.* In geometrical terms, Property 5.1 states that the boundary of the basin of attraction surrounding each pattern becomes a sharp $(N-1)$-dimensional hypersphere with radius $r$ in the limit $\beta \to \infty$. For lower, finite $\beta$, the spherical boundary becomes increasingly fuzzy. From the perspective of an energy landscape, each pattern lies in an $N$-dimensional energy minimum with infinitely steep walls when $\beta \to \infty$. As $\beta$ is lowered, the barriers become progressively smoother.

**Property 5.2** (Convergence in one step). *Given a set of unique patterns $\{\boldsymbol{\xi}^\mu\}_{\mu=1}^M$ with $\min_{\mu,\nu\neq\mu}\|\boldsymbol{\xi}^\mu - \boldsymbol{\xi}^\nu\|_2 > 2r$, the $\text{Exp}_\beta$ network at $\beta \to \infty$, initialized at $\mathbf{s}^{(0)} = \boldsymbol{\xi}^\mu + \Delta\boldsymbol{\xi}$, will converge to $\boldsymbol{\xi}^\mu$ in one step if $\|\Delta\boldsymbol{\xi}\|_2 < r$.*

**Property 5.3** (No spurious attractors). *Given a set of unique patterns $\{\boldsymbol{\xi}^\mu\}_{\mu=1}^M$ with $\min_{\mu,\nu\neq\mu}\|\boldsymbol{\xi}^\mu - \boldsymbol{\xi}^\nu\|_2 > 2r$ and $\nexists\mu : \|\boldsymbol{\xi}^\mu\|_2 = r/(1 - e^{-1})$, the only attractors of the dynamics of the $\text{Exp}_\beta$ network at $\beta \to \infty$ are the points $\{\boldsymbol{\xi}^\mu\}_{\mu=1}^M$, together with $\mathbf{0}$ if $\nexists\mu : \|\boldsymbol{\xi}^\mu\|_2 \leq r$.*

*Remark.* Properties 5.2 and 5.3 can be shown to be true simply by inserting the expression $\mathbf{s}^{(0)} = \boldsymbol{\xi}^\mu + \Delta\boldsymbol{\xi}$ in Eq. 22. Assuming no overlaps between the basins of attraction, a quick calculation shows that $\mathbf{s}^{(1)} = \boldsymbol{\xi}^\mu$ if $\|\Delta\boldsymbol{\xi}\|_2 < r$. If, on the other hand, the network is initialized such that $\|\mathbf{s}^{(0)} - \boldsymbol{\xi}^\mu\|_2 > r, \forall\mu$, one always obtains $\mathbf{s}^{(2)} = \boldsymbol{\xi}^0$, where $\boldsymbol{\xi}^0$ is either $\mathbf{0}$ or the pattern closest to $\mathbf{0}$. In other words, the network recalls a pattern only if the initialization is close enough to it. If located far from *all* patterns, the network assumes an "agnostic" state, represented either by the origin or the pattern closest to the origin (if the origin happens to be located within a basin of attraction).

In the following two properties, we evaluate how the radius of attraction $r$ determines the maximum input noise tolerance and storage capacity.

**Property 6** (Robustness to white noise). *Assume that we are given a set of unique patterns $\boldsymbol{\xi}^1, \ldots, \boldsymbol{\xi}^M \sim \mathcal{N}(\mathbf{0}, \mathbf{I}_N)$ with $\min_{\mu,\nu\neq\mu}\|\boldsymbol{\xi}^\mu - \boldsymbol{\xi}^\nu\|_2 > 2r$, and that the $\text{Exp}_\beta$ network is initialized in a distorted pattern $\mathbf{s}^{(0)} = \boldsymbol{\xi}^\mu + \boldsymbol{\epsilon}$, where $\boldsymbol{\epsilon} \sim \mathcal{N}(\mathbf{0}, \sigma^2\mathbf{I}_N)$. Then, at $\beta \to \infty$, the maximum noise variance $\sigma_{\max}^2$ with which $\boldsymbol{\xi}^\mu$ can be recovered in at least 50% of trials is*

$$\sigma_{\max}^2 = r^2/N . \tag{23}$$

**Property 7** (Exponential storage capacity). *At $\beta \to \infty$, and for $N \gg 1$, the average maximum number of patterns sampled from $\mathcal{N}(\mathbf{0}, \mathbf{I}_N)$ that the $\text{Exp}_\beta$ network can store and recall without errors is lower-bounded according to*

$$M_{\max} \geq \sqrt{2\sqrt{\pi N}(1 - 2\sigma_{\max}^2)}\exp\left[\frac{N(1 - 2\sigma_{\max}^2)^2}{8}\right] \tag{24}$$

*where $\sigma_{\max}^2$ is the maximum white noise variance tolerated by the network.*

*Remark.* Proofs can be found in Appendices E.2.2 and E.2.3. Note that Property 7 is valid in the range $\sigma_{\max}^2 \lesssim 1/2$. While the bounds are fairly tight at the upper end of the range, they become loose when $\sigma_{\max}^2 \to 0$. In this limit, which is equivalent to $r \to 0$, the storage capacity tends to infinity, as the risk of interference between patterns vanishes when their radius of attraction becomes infinitesimal.

**Comparison to the softmax network.** If patterns are randomly placed on a hypersphere instead of being normally distributed, the state update rule in Eq. 22 reduces to the form $\mathbf{s}^{(t+1)} = \mathbf{X}\,\Theta(\mathbf{X}^\top \mathbf{s}^{(t)} - \theta)$, where $\theta$ is a fixed threshold. While the capacity remains exponential (see Appendix E.3.1), the basin of attraction surrounding each pattern now forms a spherical cap instead of a ball. We can compare this to the softmax network at zero temperature, given by $\mathbf{s}^{(t+1)} = \lim_{\beta \to \infty} \mathbf{X}\,\mathrm{softmax}(\beta\mathbf{X}^\top \mathbf{s}^{(t)}) = \mathbf{X}\arg\max(\mathbf{X}^\top \mathbf{s}^{(t)})$. This model differs from the $\mathrm{Exp}_\beta$ only in a replacement of $\Theta$ with $\arg\max$. This changes the shape of the attractor basins from spherical caps to Voronoi cells, which parcellate the entire surface of the hypersphere into a Voronoi diagram (see Fig. 2). The boundary of each basin is now no longer radially symmetric around a pattern, but instead extends as far as possible in all directions. Consequently, at $\beta \to \infty$, the softmax network has larger attractor basins and always converges to one of the stored patterns, regardless of the initialization point (assuming this is not precisely on a boundary). In contrast, the $\mathrm{Exp}_\beta$ network may converge to the origin if initialized far from all patterns. This can be interpreted as an agnostic response, which indicates that the model cannot associate the input query with any of its stored patterns.

# 5 Discussion

**Biological interpretation.** Kernel memory networks can be mapped to the anatomical properties of biological neurons. Consider an individual neuron in the feature form of the recurrent network (Eq. 9). The state of neighboring neurons $\mathbf{s}$ is first transformed through $\phi(\mathbf{s})$ and thereafter projected to the neuron through the weight matrix $(\mathbf{A} \odot \mathbf{X})\phi(\mathbf{X})^\top$. When the kernel is polynomial of degree $p$, so that $K(\mathbf{x}_i, \mathbf{x}_j) = (\mathbf{x}_i^\top \mathbf{x}_j + 1)^p$, the transformation $\phi(\mathbf{s})$ consists of all elements in $\mathbf{s}$ and their cross-terms, up to degree $p$. The input to each neuron, in other words, consists of the states of all other neurons, as well as all possible combinations of their multiplicative interactions. This neuron model can be viewed as a generalized form of, for example, the multiconnected neuron [49], the clusteron [43], or the sigma-pi unit [58, p. 73]. These are all perceptrons that include multiplicative input interactions as a means to model synaptic cross-talk and cluster-sensitivity on non-linear dendrites [55] (see Fig. 1).

In the kernel form (Eq. 10), each neuron is, again, implicitly comprised of a two-stage process, whereby the raw input $\mathbf{s}$ is first transformed through the function $K(\mathbf{X}, \mathbf{s})$ and then projected through the weight matrix $\mathbf{A} \odot \mathbf{X}$. For any inner-product kernel $K = k(\mathbf{x}_i^\top \mathbf{x}_j)$, this representation can be directly identified as a two-layer neural network, where the hidden layer is defined by the weights $\mathbf{X}$ and the activation function $k$. This interpretation of the recurrent network was recently proposed in [35, 36] and discussed in relation to hippocampal-cortical interactions involved in memory storage and recall; it is particularly reminiscent of the hippocampal indexing theory [6, 61]. However, the kernel form can also be viewed as a network in which each *individual* neuron is a generalized form of the two-layered pyramidal cell model [53, 54]. This was originally proposed as an abstract neuron model augmented with non-linear dendritic processing [41]. It should be noted, however, that the idea of interpreting kernel methods as neural networks has a longer history, and has been extensively analyzed in the case of, for example, radial basis functions [51, 52]. For further details, see Appendix F.

**Summary.** We have shown that conventional kernel methods can be used to derive the weights for hetero- and auto-associative memory networks storing binary or continuous-valued patterns with maximal noise tolerance. The result is a family of optimal memory models, which we call *kernel memory networks*, which includes the SDM and MHN as special cases. This unifying framework facilitates an intuitive understanding of the storage capacity of memory models and offers new ways to biologically interpret these in terms of non-linear dendritic integration. This work formalizes the links between kernel methods, attractor networks, and models of dendritic processing.

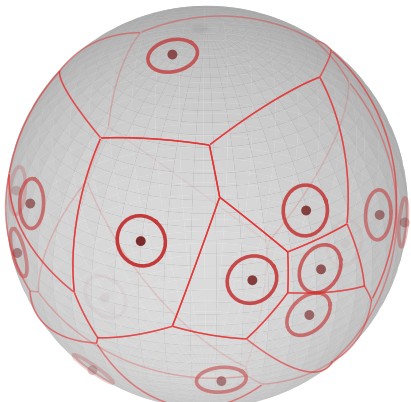

Figure 2: Plot of random patterns on $\mathbb{S}^2$ together with attractor basins at $\beta \to \infty$. Dots represent patterns ($M = 17$) while thick and thin red lines correspond to the boundaries of the attractor basins according to the $\mathrm{Exp}_\beta$ network and the softmax network, respectively. The radius of the circular boundaries has been set to half the minimum pairwise distance between the patterns.

**Future work.** A unifying theoretical framework for memory modeling can be useful for the development of both improved bio-plausible memory models and for machine learning applications. First, recognizing that there exists algorithms for training optimally noise-robust classifiers and adapting these to biological constraints can aid in the development of normative synaptic three-factor learning rules [23]. Second, the theoretical link between neuron models, kernel functions, and storage capacity enables one to fit kernel memory networks to neurophysiological data and to analyze the computational properties of biophysically informed memory models. Finally, our unifying framework reveals that most memory models differ only in the choice of kernel (model complexity) and Lagrange parameters (model precision). This categorization simplifies the tailoring of memory models to their application, and allows for the design of models whose properties fundamentally can depart from kernel memory networks, by, for example, choosing kernels not associated with a reproducing kernel Hilbert space.

## Acknowledgments and Disclosure of Funding

This study was supported by funding from the Swiss government's ETH Board of the Swiss Federal Institutes of Technology to the Blue Brain Project, a research center of the École Polytechnique Fédérale de Lausanne (EPFL).

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
