# Appendices

## A  Derivation of storage capacity scaling

### A.1  Optimal storage: The scaling of effective input dimensionality

Suppose that the kernel is a homogeneous polynomial of degree $p \ll N$, meaning $K(\mathbf{x}_i, \mathbf{x}_j) = (\mathbf{x}_i^\top \mathbf{x}_j)^p$. This implies that the associated feature map $\phi(\mathbf{x})$ contains all monomials of degree $p$ composed of the entries in $\mathbf{x}$. Moreover, given that each entry $x_i$ is $\pm 1$, each monomial in $\phi$ can be written as an interaction term of the form $x_1^{p_1} x_2^{p_2} \cdots x_N^{p_N}$, where $p_i \in \{0,1\}$ and $\sum_i^N p_i \leq p$ (i.e., no factor $x_i$ has an exponent higher than 1 and the sum of exponents is $\leq p$). The reason for this is that

$$x_i^{n_i} = \begin{cases} 1, & \text{if } n_i \text{ even} \\ x_i, & \text{if } n_i \text{ odd}. \end{cases} \tag{25}$$

The number of unique interaction terms of precisely degree $p$ (the highest degree) is $\binom{N}{p}$. As the binomial coefficient is known to be bounded according to

$$\left(\frac{N}{p}\right)^p \leq \binom{N}{p} \leq \left(\frac{Ne}{p}\right)^p \tag{26}$$

we obtain $\binom{N}{p} \sim \mathcal{O}(N^p)$, for $p$ fixed. Thus, the effective dimensionality of $\phi$ scales like $\mathcal{O}(N^p)$.

For the exponential kernel $K(\mathbf{x}_i, \mathbf{x}_j) = \exp(\mathbf{x}_i^\top \mathbf{x}_j) = \sum_{p=0}^\infty (\mathbf{x}_i^\top \mathbf{x}_j)^p / p!$, we first note that the monomials in $\phi$ now will be interaction terms of all degrees $p = 0, \ldots, N$. No monomial of degree $p > N$ will be possible. The total number of unique interaction terms will therefore be

$$\sum_{p=0}^N \binom{N}{p} = 2^N = e^{N \log 2} \tag{27}$$

where the first equality can be found in [7, p. 14]. This gives us an effective dimensionality of $\phi$ that scales like $\mathcal{O}(e^N)$.

### A.2  One-shot storage: The scaling of the support vector proliferation regime

As shown recently in [4], support vector proliferation for an SVM trained on $M$ random patterns drawn uniformly from $\{\pm 1\}^N$ occurs in the regime $N \gtrsim 2M \log M$. Solving for $M$ gives us the scaling

$$M \lesssim \frac{N}{2W_0(\frac{N}{2})} \tag{28}$$

where $W_0$ is the principal branch of the Lambert function. The largest number of patterns that can be stored in this regime is thus

$$M_{\max} \approx \frac{N}{2W_0(\frac{N}{2})}. \tag{29}$$

Using the property $W_0(x) = \log x - \log \log x + o(1)$, we can write

$$W_0\left(\frac{N}{2}\right) \sim \mathcal{O}(\log N) \tag{30}$$

which yields

$$M_{\max} \sim \mathcal{O}\left(\frac{N}{2 \log N}\right). \tag{31}$$

# B Kernel of an infinite SDM on the hypersphere

## B.1 Derivation of Eq. 17.

We follow the same steps as [8], with additional simplifications towards the end. As stated in the main text, we seek to calculate the overlapping area of two hyperspherical caps. A formula for this can be found in [38], and can be written as

$$A_\cap = A_\triangledown(R, \alpha_{\min}, \alpha_2) + A_\triangledown(R, \alpha_v - \alpha_{\min}, \alpha_1) \tag{32}$$

where

$$A_\triangledown(R, \alpha_{\min}, \alpha_2) = \frac{\pi^{\frac{N_{\mathrm{in}}-1}{2}}}{\Gamma(\frac{N_{\mathrm{in}}-1}{2})} R^{N_{\mathrm{in}}-1} \int_{\alpha_{\min}}^{\alpha_2} \sin(\varphi)^{N_{\mathrm{in}}-2} I_{1-\frac{\tan^2(\alpha_{\min})}{\tan^2(\varphi)}} \left[\frac{N_{\mathrm{in}}-2}{2}, \frac{1}{2}\right] \mathrm{d}\varphi \tag{33}$$

where $R$ is the radius, $I$ is the regularized incomplete Beta function, and

$$\alpha_1 = \alpha_2 = \arccos(b) \tag{34}$$

$$\alpha_v = \arccos(\mathbf{x}_i^\top \mathbf{x}_j) \tag{35}$$

$$\alpha_{\min} = \arctan\left(\frac{\cos(\alpha_1)}{\cos(\alpha_2)\sin(\alpha_v)} - \frac{1}{\tan(\alpha_v)}\right) \tag{36}$$

$$R = 1 . \tag{37}$$

We insert Eq. 34 in 36 and obtain

$$\alpha_{\min} = \arctan\left(\frac{1}{\sin(\alpha_v)} - \frac{\cos(\alpha_v)}{\sin(\alpha_v)}\right) = \arctan\left(\tan\left(\frac{\alpha_v}{2}\right)\right) = \frac{\alpha_v}{2} \tag{38}$$

where we have used $\tan(\alpha/2) = (1 - \cos(\alpha))/\sin(\alpha)$. Eq. 38 also follows from the symmetry of the problem. This result yields

$$A_\cap = A_\triangledown(R, \alpha_{\min}, \alpha_2) + A_\triangledown(R, \alpha_v - \alpha_{\min}, \alpha_1) = 2A_\triangledown(R, \alpha_{\min}, \alpha_1) . \tag{39}$$

We rewrite the regularized incomplete Beta function as

$$I_{1-\frac{\tan^2(\alpha_{\min})}{\tan^2(\varphi)}} \left[\frac{N_{\mathrm{in}}-2}{2}, \frac{1}{2}\right] = \frac{\Gamma(\frac{N_{\mathrm{in}}-1}{2})}{\Gamma(\frac{N_{\mathrm{in}}-2}{2})\Gamma(\frac{1}{2})} B\left[1 - \frac{\tan^2(\alpha_{\min})}{\tan^2(\varphi)}; \frac{N_{\mathrm{in}}-2}{2}, \frac{1}{2}\right] \tag{40}$$

where $B$ is the incomplete Beta function. The area of an $(N_{\mathrm{in}} - 1)$-dimensional hypersphere is

$$A_\circ = \frac{2\pi^{\frac{N_{\mathrm{in}}}{2}}}{\Gamma(\frac{N_{\mathrm{in}}}{2})} R^{N_{\mathrm{in}}-1} . \tag{41}$$

We insert Eq. 40 in 33 and use the result in Eq. 39. Using the notation $\alpha_b = \alpha_1$ and $\alpha_x = \alpha_{\min}$, and the identities

$$\frac{\Gamma(\frac{N_{\mathrm{in}}}{2})}{\Gamma(\frac{N_{\mathrm{in}}-2}{2})} = \frac{N_{\mathrm{in}}-2}{2}, \quad \Gamma\left(\frac{1}{2}\right) = \sqrt{\pi} , \tag{42}$$

the ratio between the overlapping area of the hyperspherical caps and the complete are of the hypersphere can now be calculated as

$$\frac{A_\cap}{A_\circ} = \frac{N_{\mathrm{in}}-2}{2\pi} \int_{\alpha_x}^{\alpha_b} \sin(\varphi)^{N_{\mathrm{in}}-2} B\left[1 - \frac{\tan^2(\alpha_x)}{\tan^2(\varphi)}; \frac{N_{\mathrm{in}}-2}{2}, \frac{1}{2}\right] \mathrm{d}\varphi . \tag{43}$$

## B.2 Derivation of Eq. 18.

For a large bias $b \gtrsim 0.9$, which is equivalent to a small angle $\alpha_b = \arccos(b)$, the hyperspherical caps surrounding $\mathbf{x}_i$ and $\mathbf{x}_j$ will be very small in relation to the whole hypersphere. In this case, we can neglect the curvature of the hyperspherical surface and project the area of the hyperspherical cap to the plane that cuts through the rims of the cap. This projection is a $(N_{\mathrm{in}} - 1)$-dimensional hyperball

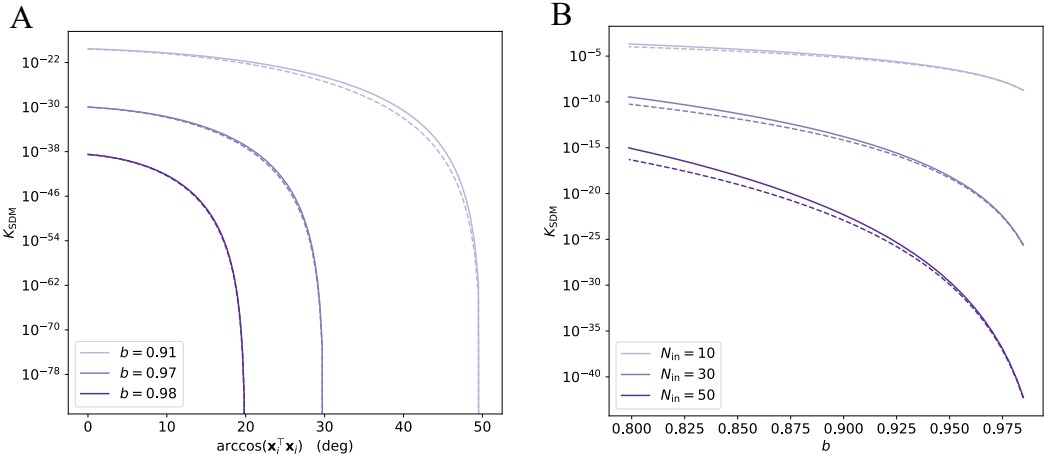

Figure B.1: Plot of the kernel of an infinite SDM on the hypersphere, $K_{\mathrm{SDM}}(\mathbf{x}_i, \mathbf{x}_j)$, as a function of (A) the angle between $\mathbf{x}_i$ and $\mathbf{x}_j$, and (B) the bias $b$. Solid lines represent the exact solution in Eq. 43, and dashed lines the approximation in Eq. 48. Parameter values: (A) $N_{\mathrm{in}} = 50$; (B) $\arccos(\mathbf{x}_i^\top \mathbf{x}_j) = \arccos(b)$.

(we will refer to it as a mini-ball). In three dimensions, for example, the projection of a spherical cap to the plane constitutes a disk, which is a 2-dimensional ball. The radius of each mini-ball is

$$\hat{b} = \sin(\arccos(b)) \tag{44}$$

and the half-distance between the centers of the mini-balls is

$$\Delta = \frac{1}{2}\|\mathbf{x}_i - \mathbf{x}_j\|_2 . \tag{45}$$

We estimate the overlapping area of the hyperspherical caps by calculating the overlapping volume of the mini-balls in $(N_{\mathrm{in}} - 1)$ dimensions. A formula for the overlapping volume of two hyperballs can be found in [39] and is

$$V_\cap = \frac{\pi^{\frac{N_{\mathrm{in}}-1}{2}}}{\Gamma(\frac{N_{\mathrm{in}}+1}{2})} \hat{b}^{N_{\mathrm{in}}-1} I_{1-\left(\frac{\Delta}{\hat{b}}\right)^2}\left[\frac{N_{\mathrm{in}}}{2}, \frac{1}{2}\right] . \tag{46}$$

We rewrite the regularized incomplete Beta function as

$$I_{1-\left(\frac{\Delta}{2\hat{b}}\right)^2}\left[\frac{N_{\mathrm{in}}}{2}, \frac{1}{2}\right] = \frac{\Gamma(\frac{N_{\mathrm{in}}+1}{2})}{\Gamma(\frac{N_{\mathrm{in}}}{2})\Gamma(\frac{1}{2})} B\left[1-\left(\frac{\Delta}{\hat{b}}\right)^2; \frac{N_{\mathrm{in}}}{2}, \frac{1}{2}\right] \tag{47}$$

and insert Eq. 47 in 46. The ratio between the overlapping area of the hyperspherical caps and the complete area of the hypersphere can now be estimated as

$$\frac{A_\cap}{A_\circ} \approx \frac{V_\cap}{A_\circ} = \frac{\hat{b}^{N_{\mathrm{in}}-1}}{2\pi} B\left[1-\left(\frac{\Delta}{\hat{b}}\right)^2; \frac{N_{\mathrm{in}}}{2}, \frac{1}{2}\right] . \tag{48}$$

A comparison of the exact solution in Eq. 43 and the approximation in Eq. 48 can be seen in Fig. B.1.

## C  Iterative learning in an SVM network

We will in this section compare the noise tolerance of a single neuron in an SVM network when trained with an iterative learning rule, and when configured according to the MHN. First, we choose to equip the neuron with the feature map $\phi_{\mathrm{pairs}}(\mathbf{x})$, which consists of all unique pairs of cross-terms $x_i x_j$, $i \neq j$. This yields a storage capacity scaling of $\mathcal{O}(N^2)$, and we therefore parameterize the storage load as $M/N^2$. We train the weights of the neuron either with the stochastic batch perceptron

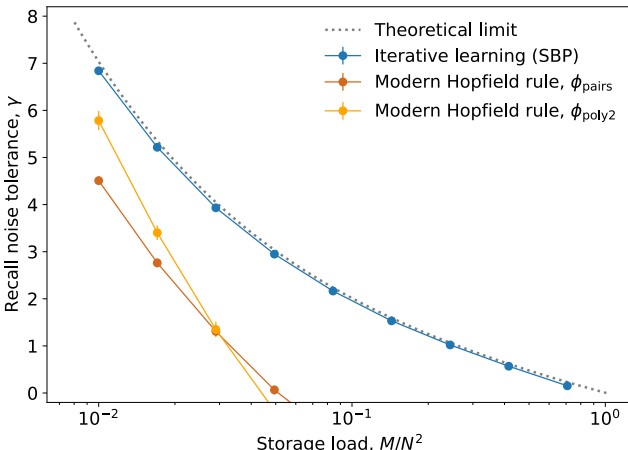

Figure C.1: Plot of the noise tolerance $\gamma$ (mean $\pm$ s.e.m. over 20 simulations) as a function of the storage load for a single SVM neuron with $N = 10^2$ inputs, trained with the stochastic batch perceptron (SBP) and the modern Hopfield rule. The SBP uses the feature map $\phi_{\text{pairs}}$, while the modern Hopfield rule is applied to both $\phi_{\text{pairs}}$ and $\phi_{\text{poly2}}$, corresponding to the kernel $K(\mathbf{x}_i, \mathbf{x}_j) = (\mathbf{x}_i^\top \mathbf{x}_j)^2$. SBP hyperparameters: learning rate $= 10^{-5}$, iterations $= 20M$.

rule [14] or with the one-shot learning rule of the MHN, which is obtained by setting $\alpha^\mu = 1, \forall \mu$, in Eq. 4, that is

$$\mathbf{w} = \sum_{\mu}^{M} \xi_{\text{out}}^\mu \phi(\boldsymbol{\xi}_{\text{in}}^\mu) \ . \tag{49}$$

Finally, we quantify the noise tolerance as the smallest Euclidean distance between the neuron's decision boundary and the patterns $\{\boldsymbol{\xi}_{\text{in}}^\mu\}_{\mu=1}^M$. This is equivalent to the minimum classification margin, defined as

$$\gamma = \min_{\mu} \frac{\xi_{\text{out}}^\mu (\mathbf{w}^\top \boldsymbol{\xi}_{\text{in}}^\mu)}{\|\mathbf{w}\|_2} \ . \tag{50}$$

We are only interested in the performance regime where all patterns are correctly recalled (i.e., correctly classified). This means that we only compare positive margins, since a negative margin indicates that there is one or more patterns that no longer can be correctly recalled. The results are plotted in Fig. C.1, and demonstrate that the margin for the MHN quickly drops with increasing load, while the iterative learning rule achieves a margin close to the theoretical optimum derived by Gardner [22]. Moreover, as the maximum storage capacity $M_{\text{max}}$ of each learning rule can be found at the intersection between the margin curve and the line $\gamma = 0$, the capacity of the online rule can be estimated to $\sim 0.7 N^2$, which is more than an order of magnitude higher than that of the MHN, which is $\sim 0.05 N^2$.

## D   Generalized pseudoinverse rule

When the network is linear and underdetermined, meaning $M < N$, we can make sure that all patterns are attractors by modeling each neuron as a *least-squares SVM* [60] instead of a conventional SVM, so that the weights satisfy

$$\min_{\mathbf{w}} \|\mathbf{w}_i\|_2 \quad \text{s.t.} \quad \mathbf{w}_i^\top \boldsymbol{\xi}^\mu = \xi_i^\mu, \quad \forall \mu, i \ . \tag{51}$$

This is a minimum-norm interpolation problem, and yields the solution

$$\mathbf{s}^{(t+1)} = \text{sgn}\left[\mathbf{X}\mathbf{K}^\dagger \mathbf{X}^\top \mathbf{s}^{(t)}\right] = \text{sgn}\left[\mathbf{X}\mathbf{X}^\dagger \mathbf{s}^{(t)}\right] \tag{52}$$

where $\mathbf{K} = \mathbf{X}^\top \mathbf{X}$ is the *kernel matrix* and $\mathbf{K}^\dagger$ its Moore-Penrose pseudoinverse, and where we have used the property $\mathbf{K}^\dagger \mathbf{X}^\top = (\mathbf{X}^\top \mathbf{X})^\dagger \mathbf{X}^\top = \mathbf{X}^\dagger$. This is the *pseudoinverse learning rule* [50].

The derivation can be extended to MHNs by performing interpolation on the feature map $\phi(\boldsymbol{\xi}^\mu)$. Assuming that the problem is still underdetermined, so that $M < N_\phi$, we aim to find the weights

$$\min_{\mathbf{w}} \|\mathbf{w}_i\|_2 \quad \text{s. t.} \quad \mathbf{w}_i^\top \phi(\boldsymbol{\xi}^\mu) = \xi_i^\mu, \quad \forall \mu, i \tag{53}$$

which, analogously to the linear case, produces the optimal state update

$$\mathbf{s}^{(t+1)} = \text{sgn}\left[\mathbf{X}\mathbf{K}^\dagger K(\mathbf{X}, \mathbf{s}^{(t)})\right] \tag{54}$$

where $\mathbf{K} = K(\mathbf{X}, \mathbf{X}) = \phi(\mathbf{X})^\top \phi(\mathbf{X})$. This can, again, be simplified to

$$\mathbf{s}^{(t+1)} = \text{sgn}\left[\mathbf{X}\phi(\mathbf{X})^\dagger \phi(\mathbf{s}^{(t)})\right] \tag{55}$$

where we can identify the weight matrix $\mathbf{W} = \mathbf{X}\phi(\mathbf{X})^\dagger$. This is the *generalized pseudoinverse learning rule*. Note that, if the feature-expanded patterns $\{\phi(\boldsymbol{\xi}^\mu)\}_{\mu=1}^M$ are linearly independent, the kernel matrix is invertible and we have $\mathbf{K}^\dagger = \mathbf{K}^{-1}$.

# E   The kernel memory network for continuous patterns

## E.1   Minimum norm interpolation and attractor basin size

**Proof of Property 4.**   In the most general variant of this setting, each neuron $i$ is modeled as a linear regressor with a neuron-specific feature map $\phi_i$ and a state $s_i$ which is updated according to

$$s_i^{(t+1)} = \mathbf{w}_i^\top \phi_i(\mathbf{s}^{(t)}) . \tag{56}$$

All patterns $\mathbf{X}$ are guaranteed to be fixed points of the dynamics by finding the weights that satisfy

$$\xi_i^\mu = \mathbf{w}_i^\top \phi_i(\boldsymbol{\xi}^\mu), \ \forall \mu, i . \tag{57}$$

In order for each pattern to also be an attractor, the weights must satisfy the additional constraint

$$\|\mathbf{J}_s\|_2 \Big|_{\mathbf{s}^{(t)} = \boldsymbol{\xi}^\mu} < 1, \ \forall \mu \tag{58}$$

where $\mathbf{J}_s$ is the Jacobian of the state update rule with respect to the input $\mathbf{s}^{(t)}$. The meaning of Eq. 58 is that the spectral norm of the Jacobian must be less than 1 when evaluated at each pattern. The reason for this is that the update rule, which computes $\mathbf{s}^{(t+1)}$ as a function of $\mathbf{s}^{(t)}$ (either synchronously or asynchronously) is continuously differentiable with respect to $\mathbf{s}^{(t)}$ and therefore satisfies the mean value inequality, so that

$$\left\|\mathbf{s}_1^{(t+1)} - \mathbf{s}_2^{(t+1)}\right\|_2 \le \hat{J}_s \left\|\mathbf{s}_1^{(t)} - \mathbf{s}_2^{(t)}\right\|_2 \tag{59}$$

where $\hat{J}_s$ is an upper bound of the spectral norm, meaning

$$\|\mathbf{J}_s\|_2 \le \hat{J}_s . \tag{60}$$

If $\hat{J}_s < 1$ at a pattern $\boldsymbol{\xi}^\mu$, it is also possible to find a neighborhood around $\boldsymbol{\xi}^\mu$ where $\hat{J}_s < 1$ holds as well, due to the continuity of the state update rule. Given this, the Banach fixed point theorem ensures that the state update rule is a contractive map in a region surrounding $\boldsymbol{\xi}^\mu$ and, equivalently, that $\boldsymbol{\xi}^\mu$ is a stable attractor [56]. While the complete basin of attraction of $\boldsymbol{\xi}^\mu$ might be difficult to compute exactly, we can define a subset of the basin as the set of points $\mathcal{S}^\mu$ in the open neighborhood of $\boldsymbol{\xi}^\mu$ satisfying

$$\mathcal{S}^\mu = \{\mathbf{s}^{(t)} : \|\mathbf{J}_s\|_2 < 1\} . \tag{61}$$

Given that the spectral norm of the Jacobian is upper bounded by the Frobenius norm, that is

$$\|\mathbf{J}_s\|_2 \le \|\mathbf{J}_s\|_F , \tag{62}$$

we can obtain a lower bound of the extent of the basin of attraction with the set

$$\widehat{\mathcal{S}}^\mu = \{\mathbf{s}^{(t)} : \|\mathbf{J}_s\|_F < 1\} . \tag{63}$$

We can write $\mathbf{J}_s$ as

$$\mathbf{J}_s = \overline{\mathbf{W}} \cdot \overline{\mathbf{J}}_\phi \tag{64}$$

where

$$\overline{\mathbf{W}} = \begin{pmatrix} \mathbf{w}_1^\top & 0 & \cdots & 0 \\ 0 & \mathbf{w}_2^\top & \cdots & 0 \\ \vdots & \vdots & \ddots & \vdots \\ 0 & 0 & \cdots & \mathbf{w}_N^\top \end{pmatrix} , \quad \overline{\mathbf{J}}_\phi = \begin{pmatrix} \mathbf{J}_{\phi_1} \\ \mathbf{J}_{\phi_2} \\ \vdots \\ \mathbf{J}_{\phi_N} \end{pmatrix} \tag{65}$$

and where $\mathbf{J}_{\phi_i}$ is the Jacobian of $\phi_i(\mathbf{s}^{(t)})$ with respect to $\mathbf{s}^{(t)}$. This gives us

$$\|\mathbf{J}_s\|_F = \|\overline{\mathbf{W}} \cdot \overline{\mathbf{J}}_\phi\|_F \le \|\overline{\mathbf{W}}\|_F \cdot \|\overline{\mathbf{J}}_\phi\|_F \tag{66}$$

where the last expression is given by the Cauchy-Schwartz inequality. Since $\|\overline{\mathbf{J}}_\phi\|_F$ depends only on the kernel, which is fixed, the right-hand side of Eq. 66 can only be minimized by finding a set of weights that minimize $\|\overline{\mathbf{W}}\|_F$. By first rewriting this norm as

$$\|\overline{\mathbf{W}}\|_F^2 = \sum_i^N \|\mathbf{w}_i\|_2^2 \tag{67}$$

we see that its minimum is obtained by minimizing $\|\mathbf{w}_i\|_2, \forall i$. Combining this requirement with Eq. 57 is equivalent to performing a minimum norm interpolation, that is

$$\min_{\mathbf{w}_i} \|\mathbf{w}_i\|_2 \quad \text{s.t.} \quad \xi_i^\mu = \mathbf{w}_i^\top \phi_i(\boldsymbol{\xi}^\mu), \quad \forall \mu, i . \tag{68}$$

If we now assume, as in the binary case, that all neurons use the same feature map, so that $\phi_i = \phi$, $\forall i$, the solution can be compactly written as in Eq. 20. This maximizes a lower bound of the attractor basin size, as defined by $\mathcal{S}^\mu$, for each pattern $\boldsymbol{\xi}^\mu$. $\qquad\square$

### E.2   Normally distributed patterns

#### E.2.1   Kernel at zero temperature

**Proof of Property 5.1.**   Using the notation $\Delta = \|\boldsymbol{\xi}^\mu - \boldsymbol{\xi}^\nu\|_2$, we have that

$$\lim_{\beta \to \infty} \left(\frac{\Delta}{r}\right)^\beta = \begin{cases} 0 , & \Delta < r \\ 1 , & \Delta = r \\ \infty , & \Delta > r \end{cases} \tag{69}$$

from which it follows that

$$\lim_{\beta \to \infty} \exp\left[-\left(\frac{\Delta}{r}\right)^\beta\right] = \begin{cases} 1 , & \Delta < r \\ e^{-1} , & \Delta = r \\ 0 , & \Delta > r \end{cases} \tag{70}$$

which is equivalent to $\Theta(r - \Delta)$ with $\Theta(0) = e^{-1}$. We combine this with the assumption that the patterns are unique and that $\min_{\mu,\nu \neq \mu} \|\boldsymbol{\xi}^\mu - \boldsymbol{\xi}^\nu\|_2 > r$ and obtain

$$\lim_{\beta \to \infty} K_{\exp_\beta}(\boldsymbol{\xi}^\mu, \boldsymbol{\xi}^\nu) = \Theta(r - \|\boldsymbol{\xi}^\mu - \boldsymbol{\xi}^\nu\|_2) = \begin{cases} 1 , & \mu = \nu \\ 0 , & \mu \neq \nu \end{cases} \tag{71}$$

which can be written compactly as

$$\lim_{\beta \to \infty} \mathbf{K}_{\exp_\beta} = \mathbf{I}_M . \tag{72}$$

It directly follows that

$$\lim_{\beta \to \infty} \mathbf{K}_{\exp_\beta}^{-1} = \mathbf{I}_M^{-1} = \mathbf{I}_M \tag{73}$$

and, therefore,

$$\lim_{\beta \to \infty} \mathbf{X} \mathbf{K}_{\exp_\beta}^{-1} K_{\exp_\beta}(\mathbf{X}, \mathbf{s}^{(t)}) = \mathbf{X} \, \Theta(r^2 - \|\mathbf{X} - \mathbf{s}^{(t)}\|_2^2) . \tag{74}$$

$$\square$$

### E.2.2 Noise robustness

**Proof of Property 6.** With $\mathbf{s}^{(0)} = \boldsymbol{\xi}^\mu + \boldsymbol{\epsilon}$, we have

$$\|\boldsymbol{\xi}^\mu - \mathbf{s}^{(0)}\|_2^2 = \|\boldsymbol{\epsilon}\|_2^2 = \|\sigma\boldsymbol{\epsilon}_0\|_2^2 = \sigma^2\|\boldsymbol{\epsilon}_0\|_2^2 \tag{75}$$

where $\boldsymbol{\epsilon}_0 \sim \mathcal{N}(\mathbf{0}, \mathbf{I}_N)$, from which it follows that $\|\boldsymbol{\epsilon}_0\|_2^2$ is a random variable with a $\chi^2(N)$ distribution. According to the central limit theorem, we also have

$$\lim_{N \to \infty} \frac{\|\boldsymbol{\epsilon}_0\|_2^2 - N}{\sqrt{2N}} \sim \mathcal{N}(0, 1) \tag{76}$$

where we have used the fact that each term in $\|\boldsymbol{\epsilon}_0\|_2^2$ is $\chi^2(1)$-distributed, and has mean 1 and variance 2. We will therefore make the approximation $\|\boldsymbol{\epsilon}_0\|_2^2 \sim \mathcal{N}(N, 2N)$ for large $N$. This gives us

$$\sigma^2\|\boldsymbol{\epsilon}_0\|_2^2 \sim \mathcal{N}(\sigma^2 N, 2\sigma^4 N) . \tag{77}$$

The original pattern $\boldsymbol{\xi}^\mu$ will only be recovered if $r^2 - \|\boldsymbol{\xi}^\mu - \mathbf{s}^{(0)}\|_2^2 \geq 0$, which is satisfied in at least 50% of trials if $r^2 \geq \sigma^2 N$. The maximum variance with which this type of recovery still holds is thus

$$\sigma_{\max}^2 = r^2/N . \tag{78}$$

$\square$

### E.2.3 Storage capacity

**Proof of Property 7.** In the limit $\beta \to \infty$, the boundary of the basin of attraction surrounding each pattern is sharp. In this setting, we are guaranteed that each pattern can be recalled without errors as long as $\min_{i,j \neq i}\|\boldsymbol{\xi}^i - \boldsymbol{\xi}^j\|_2 > 2r$. We will therefore estimate the storage capacity by calculating the number of patterns, on average, that can be loaded into the network before at least two attractor basins overlap and the condition above is violated (see Fig. E.1).

We begin by observing that for two random patterns $\boldsymbol{\xi}^i, \boldsymbol{\xi}^j \sim \mathcal{N}(\mathbf{0}, \mathbf{I}_N)$, we have

$$\frac{1}{2}\|\boldsymbol{\xi}^i - \boldsymbol{\xi}^j\|_2^2 \sim \chi^2(N) \tag{79}$$

which, using the central limit theorem as in Eq. 76, can be approximated as $\mathcal{N}(N, 2N)$ for large $N$, thereby yielding

$$\|\boldsymbol{\xi}^i - \boldsymbol{\xi}^j\|_2^2 \sim \mathcal{N}(2N, 8N) . \tag{80}$$

We now assume that the squared Euclidean distance between each pair of patterns $\boldsymbol{\xi}^i, \boldsymbol{\xi}^j$ in a set of $M$ given patterns $\{\boldsymbol{\xi}^\mu\}_{\mu=1}^M$ is an independent sample of a random variable, denoted $\Delta^2$, which is distributed as in Eq. 80. This is, of course, an approximation which neglects that the pairwise distances between any set of points are inter-dependent. Nonetheless, for relatively large $N$ and $M$, the approximation accurately describes the empirical distance distribution.

Relying on this assumption, the process of drawing $M$ random patterns becomes equivalent to drawing $M(M-1)/2$ unique pairwise distances $\Delta^2$ from the distance distribution. The probability of drawing a sample $\Delta^2 \leq 4r^2$ can be calculated using the cumulative distribution function for the standard normal distribution, given by $\Phi(x) = \frac{1}{2}\operatorname{erfc}(-x)$, according to

$$\mathbb{P}(\Delta^2 \leq 4r^2) = \frac{1}{2}\operatorname{erfc}\left(\frac{N - 2r^2}{2\sqrt{N}}\right) . \tag{81}$$

The average number of samples of $\Delta^2$ one needs to draw before a sample satisfies $\Delta^2 \leq 4r^2$ is given by $\mathbb{P}(\Delta^2 \leq 4r^2)^{-1}$. This determines the maximum number of patterns that the network, on average, can store, according to

$$\frac{M_{\max}(M_{\max} - 1)}{2} = \frac{1}{\mathbb{P}(\Delta^2 \leq 4r^2)} . \tag{82}$$

We combine this expression with the approximation $M_{\max}(M_{\max} - 1) \approx M_{\max}^2$ (which holds for large $M_{\max}$) and Eq. 81, and obtain

$$M_{\max} = 2\operatorname{erfc}\left(\frac{N - 2r^2}{2\sqrt{N}}\right)^{-1/2} . \tag{83}$$

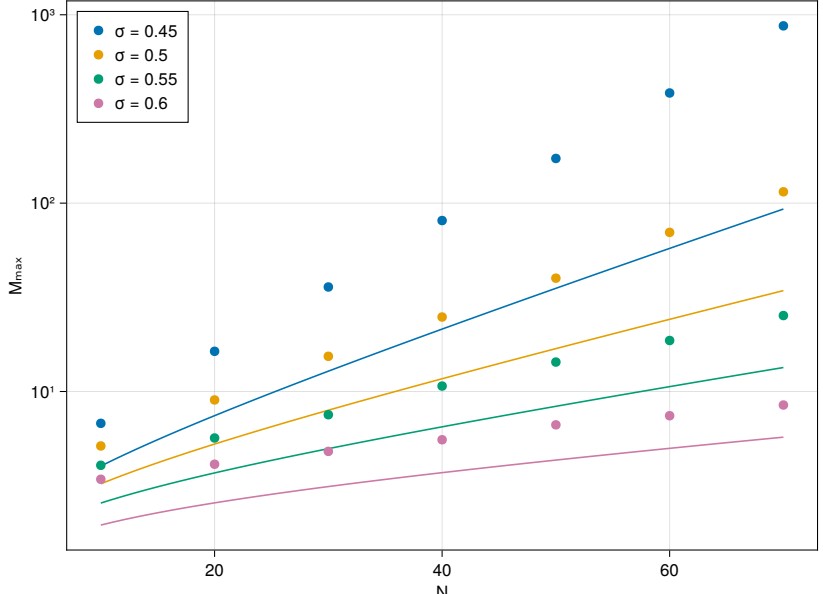

Figure E.1: Plot of the storage capacity of the $\text{Exp}_\beta$ network at $\beta \to \infty$ with normally distributed patterns. Dots represent means ($\pm$ s.e.m.) over 1000 simulations, in which the capacity is determined by the number of patterns sampled until one of the pairwise distances is smaller than $2r$. The standard error is too small to be visible. Lines correspond to the bound in Eq. 87. Note that the plot is log-linear, so the linear increase indicates that $M_{\max}$ scales exponentially in $N$.

We now parameterize the radius $r$ in terms of the largest tolerable noise amplitude, according to Eq. 78. This gives us

$$M_{\max} = 2\,\text{erfc}\left(\frac{\sqrt{N}(1 - 2\sigma_{\max}^2)}{2}\right)^{-1/2}. \tag{84}$$

Given that the $\text{erfc}$ function can be well approximated using the asymptotic expansion

$$\text{erfc}(x) \approx \frac{e^{-x^2}}{\sqrt{\pi}x} \sum_{n=0}^{\infty} (-1)^n \frac{(2n-1)!!}{(2x^2)^n} \tag{85}$$

for large arguments $x$, we can obtain a tight lower bound of $\text{erfc}^{-1}$ as long as $N$ is large and $\sigma_{\max}^2 \lesssim 1/2$ with the inverse zeroth order expansion, thereby obtaining

$$\text{erfc}(x)^{-1} \geq \sqrt{\pi}xe^{x^2}. \tag{86}$$

We insert Eq. 86 in 84 and finally obtain

$$M_{\max} \geq \sqrt{2\sqrt{\pi N}(1 - 2\sigma_{\max}^2)} \exp\left[\frac{N(1 - 2\sigma_{\max}^2)^2}{8}\right]. \tag{87}$$

$\square$

### E.3 Patterns on the hypersphere

#### E.3.1 Storage capacity

**Property 8** (Storage capacity: patterns on the hypersphere). *At $\beta \to \infty$, the average maximum number of patterns that the $\text{Exp}_\beta$ network can store and recall without errors is lower-bounded by*

$$M_{\max} \geq \sqrt{\sqrt{8\pi N}(1 - 2r^2)} \exp\left[\frac{N(1 - 2r^2)^2}{4}\right] \tag{88}$$

*when each pattern is randomly drawn from $\mathbb{S}^{N-1}$.*

*Proof.* We begin by observing that for two random patterns $\boldsymbol{\xi}^i, \boldsymbol{\xi}^j \in \mathbb{S}^{N-1}$, we have

$$\|\boldsymbol{\xi}^\mu - \boldsymbol{\xi}^\nu\|_2^2 = 2(1 - \boldsymbol{\xi}^{\mu\top}\boldsymbol{\xi}^\nu) . \tag{89}$$

The probability distribution for the inner product $\omega = \boldsymbol{\xi}^{\mu\top}\boldsymbol{\xi}^\nu$ can be found in [62], and is given by

$$\omega \sim \frac{1}{\sqrt{\pi}} \frac{\Gamma(\frac{N}{2})}{\Gamma(\frac{N-1}{2})} (1 - \omega^2)^{\frac{N-3}{2}} \tag{90}$$

which, for large $N$, can be approximated as

$$\omega \sim \mathcal{N}(0, \frac{1}{N}) . \tag{91}$$

We use Eq. 91 in 89 and obtain

$$\Delta^2 \sim \mathcal{N}(2, \frac{4}{N}) . \tag{92}$$

The probability of placing a pair of random points on $\mathbb{S}^{N-1}$ with $\Delta^2 \leq 4r^2$ is thus

$$\mathbb{P}(\Delta^2 \leq 4r^2) = \frac{1}{2} \operatorname{erfc}\left( \frac{\sqrt{N}(1 - 2r^2)}{\sqrt{2}} \right) . \tag{93}$$

The average number of samples of $\Delta^2$ one needs to draw before a sample satisfies $\Delta^2 \leq 4r^2$ is given by $\mathbb{P}(\Delta^2 \leq 4r^2)^{-1}$, and the maximum number of patterns that the network, on average, can store, is therefore

$$M_{\max} = 2\operatorname{erfc}\left( \frac{\sqrt{N}(1 - 2r^2)}{\sqrt{2}} \right)^{-1/2} . \tag{94}$$

We use the lower bound in Eq. 86 in 94 and finally obtain

$$M_{\max} \geq \sqrt{\sqrt{8\pi N}(1 - 2r^2)} \exp\left[ \frac{N(1 - 2r^2)^2}{4} \right] . \tag{95}$$

$\square$

## E.4 Bipolar patterns

### E.4.1 Noise robustness

**Property 9.1** (Robustness to flipped bits)**.** *Assume that we are given a set of unique patterns* $\boldsymbol{\xi}^1, \dots, \boldsymbol{\xi}^M \in \{\pm 1\}^N$ *with* $\min_{\mu, \nu \neq \mu} \|\boldsymbol{\xi}^\mu - \boldsymbol{\xi}^\nu\|_2 > 2r$, *and that the Exp$_\beta$ network is initialized in a distorted pattern* $\mathbf{s}^{(0)} = \boldsymbol{\xi}^\mu \odot \boldsymbol{\epsilon}$, *where* $\boldsymbol{\epsilon} \in \{\pm 1\}^N$, *with* $\mathbb{P}(\epsilon_i = -1) = \rho, \forall i$. *Then, at* $\beta \to \infty$, *the maximum bit-wise error probability* $\rho_{\max}$ *with which* $\boldsymbol{\xi}^\mu$ *can be recovered in at least 50% of trials is*

$$\rho_{\max} = r^2/4N . \tag{96}$$

*Proof.* With $\mathbf{s}^{(0)} = \boldsymbol{\xi}^\mu \odot \boldsymbol{\epsilon}$, we have

$$\|\boldsymbol{\xi}^\mu - \mathbf{s}^{(0)}\|_2^2 = \|2\boldsymbol{\epsilon}_\mathcal{B}\|_2^2 = 4\|\boldsymbol{\epsilon}_\mathcal{B}\|_2^2 \tag{97}$$

where $\boldsymbol{\epsilon}_\mathcal{B} \in \{0, 1\}^N$, with each entry being a random variable distributed as $(\epsilon_\mathcal{B})_i \sim \text{Bernoulli}(\rho)$. This implies that $\|\boldsymbol{\epsilon}_\mathcal{B}\|_2^2 \sim \text{Binomial}(N, \rho)$, which can be approximated as $\mathcal{N}(\rho N, \rho(1 - \rho)N)$ for large $N$. This gives

$$4\|\boldsymbol{\epsilon}_\mathcal{B}\|_2^2 \sim \mathcal{N}(4\rho N, 16\rho(1 - \rho)N) . \tag{98}$$

Again, the original pattern $\boldsymbol{\xi}^\mu$ will only be recovered if $r^2 - \|\boldsymbol{\xi}^\mu - \mathbf{s}^{(0)}\|_2^2 \geq 0$, which is satisfied in at least 50% of trials if $r^2 \geq 4\rho N$. The maximum bit-wise error probability with which this type of recovery still holds is thus

$$\rho_{\max} = r^2/4N . \tag{99}$$

$\square$

In Eq. 78, $\sigma$ roughly quantifies the maximum noise fluctuation around a pattern that is tolerable with a given radius $r$, while still being able to recover the pattern in a majority of trials. In the case of Eq. 99, $\rho$ instead quantifies the maximum tolerable bit-wise error probability.

### E.4.2 Storage capacity

**Property 9.2** (Storage capacity: bipolar patterns). *At $\beta \to \infty$, the average maximum number of bipolar patterns with sparseness $f$ that the $Exp_\beta$ network can store and recall without errors is lower-bounded by*

$$M_{\max} \geq 2 \left( \frac{\pi N}{2\tilde{f}(1-\tilde{f})} \right)^{1/4} \left( \tilde{f} - 4\rho_{\max} \right)^{1/2} \exp \left[ \frac{N(\tilde{f} - 4\rho_{\max})^2}{4\tilde{f}(1-\tilde{f})} \right] \tag{100}$$

*where $\tilde{f} = 2f(1-f)$ and $\rho_{\max}$ is the maximum bit-wise error probability tolerated by the network.*

*Proof.* This proof is, again, a slightly modified variant of the proof of Property 7. First, we observe that for two random patterns $\boldsymbol{\xi}^\mu, \boldsymbol{\xi}^\nu \in \{\pm 1\}^N$ with sparseness $f$, so that $\mathbb{P}(x_i^{\mu,\nu} = 1) = f$, it is true that

$$\frac{1}{4} \|\boldsymbol{\xi}^\mu - \boldsymbol{\xi}^\nu\|_2^2 \sim \mathrm{Binomial}(N, \tilde{f}) \tag{101}$$

where $\tilde{f} = 2f(1-f)$ denotes the probability that $\boldsymbol{\xi}^\mu$ and $\boldsymbol{\xi}^\nu$ differ at any given entry. For large $N$, we can again approximate $\mathrm{Binomial}(N, \tilde{f})$ with $\mathcal{N}(\tilde{f}N, \tilde{f}(1-\tilde{f})N)$, which gives us

$$\|\boldsymbol{\xi}^\mu - \boldsymbol{\xi}^\nu\|_2^2 \sim \mathcal{N}(4\tilde{f}N, 16\tilde{f}(1-\tilde{f})N) . \tag{102}$$

We use this to compute the upper bound of the probability of drawing a distance $\Delta^2$ which satisfies $\Delta^2 \leq 4r^2$, as in Eq. 81. The result is

$$\mathbb{P}(\Delta^2 \leq 4r^2) = \frac{1}{2} \mathrm{erfc} \left( \frac{\tilde{f}N - r^2}{\sqrt{2\tilde{f}(1-\tilde{f})N}} \right) . \tag{103}$$

Following the same derivations used to produce Eqs. 82 and 83, we arrive at

$$M_{\max} = 2 \, \mathrm{erfc} \left( \frac{\tilde{f}N - r^2}{\sqrt{2\tilde{f}(1-\tilde{f})N}} \right)^{-1/2} . \tag{104}$$

As before, we parameterize the radius $r$ in terms of the maximum tolerable bit-wise error probability according to Eq. 99 and yield

$$M_{\max} = 2 \, \mathrm{erfc} \left( \frac{\sqrt{N}(\tilde{f} - 4\rho_{\max})}{\sqrt{2\tilde{f}(1-\tilde{f})}} \right)^{-1/2} . \tag{105}$$

We finally replace $\mathrm{erfc}$ with the lower bound in Eq. 86. This is valid for large $N$ and when $\rho \lesssim \tilde{f}/4$. We obtain

$$M_{\max} \geq 2 \left( \frac{\pi N}{2\tilde{f}(1-\tilde{f})} \right)^{1/4} \left( \tilde{f} - 4\rho_{\max} \right)^{1/2} \exp \left[ \frac{N(\tilde{f} - 4\rho_{\max})^2}{4\tilde{f}(1-\tilde{f})} \right] . \tag{106}$$

$\square$

## F Comparison to neuron models with active dendrites

To demonstrate how single neurons in kernel memory networks are generalizations of abstract neuron models with active dendrites, we begin by considering a neuron in the feature form (Eq. 5). We will here use the Heaviside activation function with threshold $\theta$, denoted $\Theta_\theta$, instead of $\mathrm{sgn}$, and assume that patterns and states are in $\{0, 1\}^N$, where $N$ is the number of inputs. This, however, does not change the fundamental properties of our model, as SVMs can be formulated for binary data with minor modifications. Assuming a polynomial feature map $\phi$ of degree $p$, the feature vector will

consist of all possible monomials of degree $\leq p$ composed of the states of its input neurons. Setting, for example, $p = 2$ gives us

$$
\begin{aligned}
\phi(\mathbf{x}) = (1, \sqrt{2}x_1, \ldots, \sqrt{2}x_N, \\
\sqrt{2}x_1x_2, \ldots, \sqrt{2}x_1x_N, \sqrt{2}x_2x_3, \ldots, \sqrt{2}x_{N-1}x_N, \\
x_1^2, \ldots, x_N^2) \, .
\end{aligned} \tag{107}
$$

By limiting the feature map to only include a subset of all terms, our model is reduced to the 2-degree *sigma-pi unit* [58, p. 73], which can be written as

$$
s_{\text{out}} = \Theta_\theta \left[ \sum_i w_i \prod_{j \in \mathcal{C}_i} s_{\text{in},j} \right] \tag{108}
$$

where $w_i$ is the weight of cluster $i$, which consists of a product of all inputs $s_{\text{in},j}$ whose indices $j$ are contained in the set $\mathcal{C}_i$. From a neurophysiological perspective, each product represents the cross-talk between a set of synapses. By including such multiplicative interactions, synapses can both gate and amplify each other, to generate supra-linear input currents.

If we now further constrain this model to include only a subset of the cross-terms $x_i x_j$, $i \neq j$, and parameterize each cross-term weight as $w_{ij} = w_i w_j$, our neuron model is reduced to the *clusteron* [43], which can be written as

$$
\begin{aligned}
s_{\text{out}} &= \Theta_\theta \left[ \sum_i^N \sum_{j \in \mathcal{C}_i} w_i w_j s_{\text{in},i} s_{\text{in},j} \right] \\
&= \Theta_\theta \left[ \sum_i^N w_i s_{\text{in},i} \left( \sum_{j \in \mathcal{C}_i} w_j s_{\text{in},j} \right) \right]
\end{aligned} \tag{109}
$$

where $\mathcal{C}_i$ now is the set describing all inputs $j$ that input $i$ should be paired with.

We now consider a neuron in the kernel form (Eq. 6) with an inner-product kernel $K(\mathbf{x}_i, \mathbf{x}_j) = k(\mathbf{x}_i^\top \mathbf{x}_j)$. By setting $\xi_{\text{out}}^\mu = 1$, $\forall \mu$, and assuming binary inputs, our model is reduced to

$$
s_{\text{out}} = \Theta_\theta \left[ \sum_\mu^M \alpha^\mu k \left( \sum_i^N \xi_{\text{in},i}^\mu s_{\text{in},i} \right) \right] \tag{110}
$$

which is equivalent to the pyramidal cell as a 2-layered neural network, as defined in [53]. According to the original interpretation, this neuron model is comprised of $M$ subunits, which can represent, for example, separate parts of a dendritic tree. All subunits receive the inputs and produce separate outputs which are all summed in the soma. Each subunit $\mu$ is characterized by the input weights $\boldsymbol{\xi}_{\text{in}}^\mu$ (which serves as a mask), the output weight $\alpha^\mu$ (which determines how strongly the subunit influences the response at the soma), and the activation function $k$.