# OpenReview forum: "Kernel Memory Networks: A Unifying Framework for Memory Modeling"
_NeurIPS.cc/2022/Conference — NeurIPS 2022 Accept_

### Official Review · Reviewer_a96M · 2022-06-29

**Rating:** 5
**Confidence:** 2
**Soundness:** 2 fair
**Presentation:** 2 fair
**Contribution:** 3 good

**Summary:**

This work pursued a classical yet interesting problem that designing a neural network to store a set of patterns with maximal noise robustness, and presented alternative understandings of this problem by investigating several kernel attractor networks. The claimed contributions can be summarized as follows:
1. proposed two mathematical structures of feedforward and recurrent memory networks for binary patterns.
2. show that the normative models include, as special cases, the classical and modern Hopfield network, as well as the SDM.
3. derived a simple attractor network model for storing an exponential number of continuous-valued patterns with a finite basin of attraction.

**Questions:**

What is the definition or formulation of the maximal noise robustness. It is difficult to get any noise or robustness information from Property 1, 2.1, and 2.2 unless I missed something.

**Limitations:**

Nothing mentioned.

**Strengths And Weaknesses:**

Strengths:
1. This paper is organized clearly, and the authors provided many plots for illustrating the structure of the proposed networks.
2. The derivation procedure of the feedforward and recurrent memory networks are easy to follow.
3. The result about storing an exponential number of continuous-valued patterns with a finite basin of attraction seems to be a laudable performance.

Weaknesses:
Limited to my knowledge, this paper is too hard to follow and comment. Here, I list the most concerned points, which may hinder the acceptance.
1. Is there something over-claimed? For example, the discussions about its similarity to attention and new biological interpretations seem only to give some understandings from the perspectives of attention and other fields with slight analysis. It would be better to provide detailed and solid support for this claim.
2. There lacks some clear-out and comparative results between the proposed networks and the previous ones. For example, what are the advantages of this framework that the authors proposed here.

---

> ### Author Response · Authors · 2022-08-02
> **Response to reviewer a96M**
>
> We thank the reviewer for the suggested improvements. Here are some of the corrections we have made in the latest draft:
>
> 1. *Is there something over-claimed? For example, the discussions about its similarity to attention and new biological interpretations seem only to give some understanding from the perspectives of attention and other fields with slight analysis. It would be better to provide detailed and solid support for this claim.*
>
> a) To clarify our comparison between the kernel attractor network and the softmax network, we have added an extended discussion in Appendix E.5. There, the differences between the two models are explained in more detail. To illustrate these differences, we have also added a plot of stored patterns together with attractor basins for each model in the case of N=3 dimensions (Fig. E.2).
>
> b) Regarding the biological interpretation of kernel attractor networks in terms of neurons with active dendrites, we have added an extended discussion on this topic, together with a mathematical analysis, in Appendix F. There, we specifically derive the sigma-pi unit, the clusteron, and the pyramidal cell as a 2-layered network, starting from a single neuron in the kernel attractor network.
>
>
> 2. *There lacks some clear-out and comparative results between the proposed networks and the previous ones. For example, what are the advantages of this framework that the authors proposed here.*
>
> We have added a performance comparison between a kernel attractor network (trained with a numerical method) and the modern Hopfield network (MHN; a suboptimal special case). The details can be found in Appendix C. To train a kernel attractor network, one typically needs to use a numerical method (this is mentioned in line 231). Some of these techniques are listed in the section “Online Learning”. To do the comparison, we use one of these methods (the stochastic batch perceptron) and train a neuron in a kernel attractor network. We also train a neuron according to the modern Hopfield connectivity rule (this is explained in Appendix C). We quantify the recall performance as the noise robustness of the neurons, meaning the smallest Euclidean distance between the neuron’s classification boundary and the stored patterns (equation provided in Appendix C). The results are plotted in Fig. C.1, and illustrate how a neuron in a kernel attractor network (without any initial constraints) has larger storage capacity and can be trained to recall patterns with much higher noise robustness than a neuron configured according to the MHN.
>
>
> 3. *What is the definition or formulation of the maximal noise robustness. It is difficult to get any noise or robustness information from Property 1, 2.1, and 2.2 unless I missed something.*
>
> The definition of maximum noise robustness is given in line 58 and line 92. We define it as a maximal Euclidean distance between a neuron’s classification boundary and a stored pattern. By optimizing all neurons in a network to achieve maximum noise robustness, we derive Properties 1, 2.1, and 2.2 (the optimality is satisfied by these solutions). We have also added a mathematical definition of the noise robustness across many patterns in the performance comparison in Appendix C (see parameter $\gamma$). This quantity is plotted as a function of storage load in Fig. C.1.
>
>
> 4. *Weaknesses: Limited to my knowledge, this paper is too hard to follow and comment.*
>
> To make it easier to understand the graphical structure of different variants of kernel attractor networks, we have improved and extended Fig. 1. It now includes graphical representations of feed-forward and recurrent SVM networks, as well as the sparse distributed memory (SDM) and an illustration of a single SVM neuron as a pyramidal cell.

---

### Official Review · Reviewer_5xox · 2022-07-04

**Rating:** 7
**Confidence:** 4
**Soundness:** 4 excellent
**Presentation:** 3 good
**Contribution:** 3 good

**Summary:**

This work provides a nice framework to describe hetero- and auto-associative memory models. This framework can be used to describe dynamics and capacity of different existing models, such as SDMs and Hopfield networks. As a background, the authors provide an explanation to kernel methods, that are then needed to understand the contributions of the paper: they show how a kernel trick for "one hidden layer nets" is equivalent to finding the best set of weights with minimum norm, and a general property of multilevel feedforward kernels.
The first contribution (Section 3), consists in defining auto associative memory models with binary states as recurrent SVMs. Particularly, they state that the larger the attractor basins, the more robust the model. This is then used to draw connections with prowler models in the literature, such as SDMs and Hopfield models. This study is then extended to include recent models, generalized to continuous patterns.


**Questions:**

You state that your "hope is that a proper understanding of the theoretical overlap of contemporary memory models can facilitate the development" of new models. Generally, I agree that fundamental works that are able to identify underlying common principles of different models are important in the literature. However, in your specific case, how do you think your formulation will help future progress in the field? How will your work influence future applications of AM models?


Minor:

Remove one of the two remarks at lines 82 and 102





**Limitations:**

The very few limitations (it is a theoretical paper) are discussed, and I do not see any negative societal impact of this work.

**Strengths And Weaknesses:**

Pros:

 Very nice paper, well written and straight to the point: it does exactly what is stated in the title.

Cons:

I believe the paper is suffering the 9 pages limit, as the authors use most of the space to derive the results, while having a little more space would allow interesting discussions. Particularly, the introduction and conclusion do not address future works/ possible implication of the work in any way.

---

> ### Author Response · Authors · 2022-08-02
> **Response to reviewer 5xox**
>
> *However, in your specific case, how do you think your formulation will help future progress in the field? How will your work influence future applications of AM models?*
>
> For a general answer to how our work can influence future applications of associative memory models, see the response to reviewer ztkh (and the new subsection “Future work” in the Discussion). One thing to add here, as regards biologically relevant associative memory, is the fact that our work demonstrates that the choice of kernel function is equivalent to choosing a specific neuron model. While a linear kernel attractor network (like the classical Hopfield) corresponds to a circuit of perceptrons (i.e., neurons without active dendrites), a non-linear kernel attractor network corresponds to a memory model with neurons whose dendrites are active. Our framework therefore allows one to draw a link between neuron models (which can be fit to neurophysiological data), kernel functions, and the capacity of neurons to store and process information as part of a recurrent circuit. In other words, one can fit a kernel attractor network to experimental data, and thereby analyze memory models that are biophysically informed.
>
> The remark at line 82 was indeed redundant and has been removed.

---

### Official Review · Reviewer_ztkh · 2022-07-11

**Rating:** 6
**Confidence:** 3
**Soundness:** 3 good
**Presentation:** 3 good
**Contribution:** 3 good

**Summary:**

This paper is a theoretical work in which the authors derive a set of normative models that describe the general structure of memory networks that can perform error-free recall of a given set of input patterns. Starting from well-known properties of binary classifiers they show how hetero-associate and auto-associate memories can be formulated as feed-forward and recurrent SVM networks, respectively. They also characterize the storage capacity of these networks in terms of their Vapnik-Chervonenkis dimension. Importantly, they show that previously proposed memory models such as Kanerva's sparse distributed memory and Hopfield networks are special cases of the SVM networks. They also extend their auto-associate memory model to be able to store continuous-valued patterns instead of just binary-valued patterns. Finally, they discuss how the models developed here could potentially be mapped to anatomical properties of biological neurons.

**Questions:**

- Are there any ideas on how the theory developed here could give rise to better models of memory?

**Limitations:**

The authors discuss the limitations of their approach.



**Strengths And Weaknesses:**

This paper is a purely theoretical work that develops general expressions for optimal weights in memory networks storing binary or continuous-valued patterns with maximal noise tolerance. It is interesting to think about how the general theory developed here could give rise to better models of memory, and how they could lead to better AI methods.

Also, the mapping of kernel attractor networks to anatomical properties of biological neurons is potentially an interesting avenue of research.

The presentation is also very clear and it is easy to follow the main ideas.

---

> ### Author Response · Authors · 2022-08-02
> **Response to reviewer ztkh**
>
> *Are there any ideas on how the theory developed here could give rise to better models of memory?*
>
> We have extended our Discussion with a subsection called “Future work”, where we elaborate on how our work can contribute to improved memory modeling. There are primarily two ways in which we see the results in our paper as being useful in the development of new models. First, from a neuroscientific perspective, deriving mathematical optimality conditions can be useful in the development of normative biologically plausible learning rules for storing memory engrams. For example, by first recognizing that many online algorithms originally used to train SVMs can be used to train optimally noise-robust memory networks, one can adapt these algorithms to fit biological constraints and thereby design synaptic three-factor learning rules that robustly store and consolidate engrams in attractor networks.
>
> Second, from the perspective of machine learning applications, our work not only highlights that most well-known memory models share a common theoretical framework, but it also shows that these models typically differ only in the shape of their kernel (compare the modern Hopfield network (MHN) with the sparse distributed memory (SDM)). The choice of kernel has a large impact on the computational properties of a memory model (see, e.g., Appendix E.5). The behavior of a memory model can therefore be tailored to an application by choosing an appropriate kernel function, while the precision of the model can be determined by choosing (or fitting) the Lagrange parameters. Our work is, to the best of our knowledge, the first that demonstrates these general properties for memory models through a common theoretical framework. The kernel attractor framework also shows that it is possible to depart from the regime of previous memory models by choosing a kernel that is not associated with a reproducing kernel Hilbert space (see, e.g., the softmax). This allows for the possibility of designing models with very different computational properties than the SDM or the MHN.

---

> > ### Comment · Reviewer_ztkh · 2022-08-09
> > **Response to rebuttal**
> >
> > Thank you for your detailed response to my question. I agree with reviewer 5xox that this paper is constrained by the page limit, so thank you for extending the discussion section.

---

### Meta-Review · Area_Chair_HYBL · 2022-08-26

**Recommendation:** Accept
**Confidence:** Less certain

**Metareview:**

The paper is well written and, in addition to provide a unified perspective on hetero-associative and auto-associative memories via feed-forward and recurrent SVM networks, it shows how in the proposed framework is possible to use conventional kernel methods to design an attractor network that can store an exponential number of continuous-valued patterns with a finite basin of attraction. In the rebuttal/revised version of the paper the authors properly addressed the issues raised by the reviewers. Overall, given the quality of the contribution and no significant negative issue raised by the reviewers, I think the paper can constitute a valid contribution to the NeurIPS program.

**Award:**

No

---

### Decision · Program_Chairs · 2022-09-14

Accept